# PABBO: Preferential Amortized Black-Box Optimization

**Xinyu Zhang**[*], **Daolang Huang**[*], **Samuel Kaski**
Department of Computer Science
Aalto University
`xinyu.zhang@aalto.fi`

**Julien Martinelli**
Inserm Bordeaux Population Health
Vaccine Research Institute
Université de Bordeaux
Inria Bordeaux Sud-ouest

## Abstract

Preferential Bayesian Optimization (PBO) is a sample-efficient method to learn latent user utilities from preferential feedback over a pair of designs. It relies on a statistical surrogate model for the latent function, usually a Gaussian process, and an acquisition strategy to select the next candidate pair to get user feedback on. Due to the non-conjugacy of the associated likelihood, every PBO step requires a significant amount of computations with various approximate inference techniques. This computational overhead is incompatible with the way humans interact with computers, hindering the use of PBO in real-world cases. Building on the recent advances of amortized BO, we propose to circumvent this issue by fully amortizing PBO, meta-learning both the surrogate and the acquisition function. Our method comprises a novel transformer neural process architecture, trained using reinforcement learning and tailored auxiliary losses. On a benchmark composed of synthetic and real-world datasets, our method is several orders of magnitude faster than the usual Gaussian process-based strategies and often outperforms them in accuracy.

## 1 Introduction

Learning the latent utility function of a given user from its feedback is a task that is becoming increasingly important in the era of personalization, with applications ranging from the optimization of visual designs (Koyama et al., 2020), thermal comfort (Abdelrahman & Miller, 2022), proportional integral controller (Coutinho et al., 2024), or robotic gait (Li et al., 2021). As humans are typically better at comparing two options rather than assessing their absolute value (Kahneman & Tversky, 2013), current solutions often leverage data based on *preferential* feedback, the outcome of a pairwise comparison also referred to as a *duel* (Chu & Ghahramani, 2005).

Preferential Bayesian Optimization (PBO, González et al. (2017)) is the gold standard method for optimizing black-box functions from sequential duel feedback. To learn the latent utility of a human subject, PBO operates in a Bayesian framework and classically relies on a Gaussian Process (GP) prior (Rasmussen & Williams, 2006) and a probit or logit likelihood (Brochu et al., 2010). Due to the non-conjugacy of the preferential likelihood, approximation techniques are required for posterior inference (Takeno et al., 2023). The obtained posterior is then plugged into an *acquisition function*, which selects the next pair of candidate designs to display to the user, in a way that balances exploration and exploitation (Garnett, 2023). Approximating the posterior and maximizing the acquisition function results in a significant computational overhead, which may hinder the use of PBO in real-world cases, as the user would have to wait an extended period of time before each iteration.

To circumvent this issue, amortization, or pre-computing solutions and learning an ML model to approximate them, has emerged as a promising solution, effectively enabling rapid and still accurate on-line computation with sophisticated models, trading off the speed to off-line computations. Amortization has been successfully applied to simulation-based inference (Cranmer et al., 2020; Gloeckler et al., 2024), sequential Bayesian experimental design (Foster et al., 2021), and more relevantly, vanilla Bayesian optimization (Chen et al., 2017; Maraval et al., 2023; Song et al., 2024).

---

[*] Equal contribution.

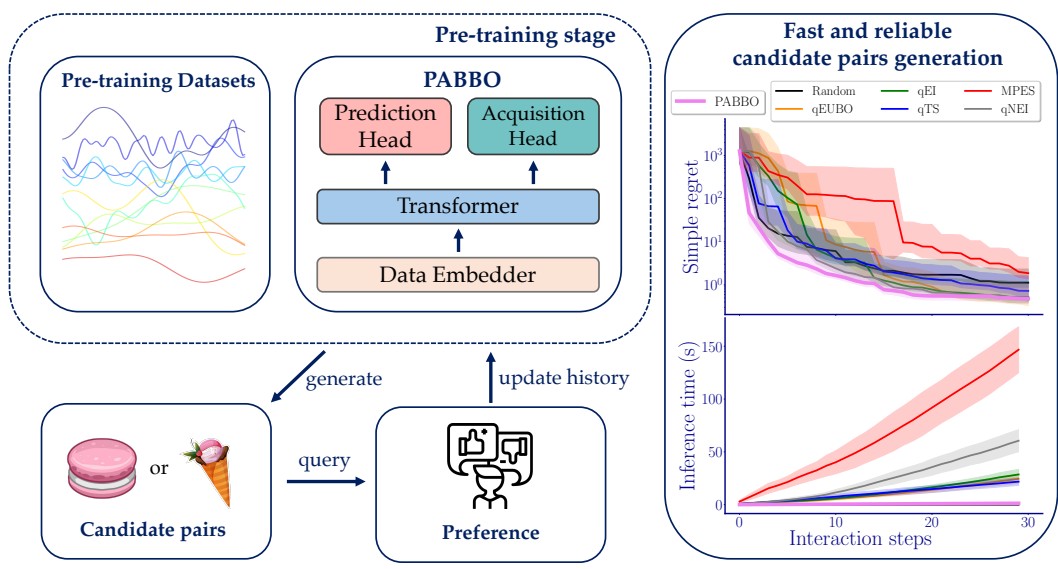

| Method | Preference Feedback | Amortization | End-to-End training | Reference |
|---|---|---|---|---|
| BO | ✗ | ✗ | ✗ | (Garnett, 2023) |
| qEUBO, qEI, qNEI qTS, MPES | ✓ | ✗ | ✗ | (Astudillo et al., 2023; Siivola et al., 2021) (Hernández-Lobato et al., 2017; Nguyen et al., 2020) |
| PFN | ✗ | ✓ | ✗ | (Müller et al., 2023) |
| NAP | ✗ | ✓ | ✓ | (Maraval et al., 2023) |
| PABBO | ✓ | ✓ | ✓ | This work |

Figure 1: Leveraging synthetic and existing BO datasets, PABBO learns an acquisition policy in an end-to-end manner, thus directly amortizing the design proposal step. Contrary to existing methods, at inference time, PABBO only relies on preference data. Our method outperforms GP-based strategies and dramatically speeds up the optimization.

Beyond speed gains, an interesting characteristic of amortization with regards to BO and PBO is that it can jointly model the probabilistic surrogate and the acquisition function, two modules that are usually treated independently in BO. This provides two powerful advantages: rather than guessing which combination of GP kernel and acquisition function works best for a given task through various heuristics, these are now learned from data. In the end, the model will directly predict the acquisition function values. However, current end-to-end black-box optimization methods predict acquisition function values for single designs. Translating this task in the preferential framework requires predicting acquisition values for pairs of designs, suggesting novel network architectures and training procedures. To the best of our knowledge, amortization has not yet been employed in a PBO setting.

**Contributions.**

- We introduce PABBO, the first end-to-end framework for Preferential Amortized Black-Box Optimization (Figure 1). PABBO directly outputs acquisition function values for candidate pairs, conditioned on observed data, and handles preferential feedback through a novel transformer-based architecture.
- We present a reinforcement learning-based pre-training scheme to directly learn acquisition function values for pairs. Due to the sparsity of the rewards, in a similar spirit as Maraval et al. (2023), we add an auxiliary loss, but based on binary classification instead of regression, due to the preferential nature of the data. Figure 1 displays the PABBO workflow.
- On a set of synthetic and real-world examples, PABBO provides speedups of several orders of magnitude against GP-based methods, often outperforming them, even on unseen examples during pre-training, thus demonstrating its impressive in-context optimization capability. We conclude by illustrating the robustness of PABBO through a series of ablation studies.

## 2 PRELIMINARIES

### 2.1 PREFERENTIAL BLACK-BOX OPTIMIZATION

The goal of preferential black-box optimization (PBBO) is to determine the optimal solution $\boldsymbol{x}^* = \arg\max_{\boldsymbol{x} \in \mathbb{X}} f(\boldsymbol{x})$ using the duel feedback $\boldsymbol{x} \succ \boldsymbol{x}'$, signifying a preference of $\boldsymbol{x}$ over $\boldsymbol{x}'$. The preference structure is governed by the latent function $f : \mathbb{X} \to \mathbb{R}$ defined over the subset $\mathbb{X} \subset \mathbb{R}^d$:

$$\boldsymbol{x} \succ \boldsymbol{x}' \iff f(\boldsymbol{x}) + \varepsilon > f(\boldsymbol{x}') + \varepsilon', \tag{1}$$

where $\varepsilon$ and $\varepsilon'$ are i.i.d. realizations of the normal distribution $\mathcal{N}(0, \sigma_{\text{noise}}^2)$.

By definition of PBBO, we only have access to a dataset $\mathcal{D}$ made of the human observations of $m$ duel feedbacks $\{l_i\}_{i=1}^m = \{\boldsymbol{x}_i \succ \boldsymbol{x}_i'\}_{i=1}^m$, with $\boldsymbol{x}_i$ and $\boldsymbol{x}_i'$ being the winner and the loser of the duel, respectively, and $l_i$ a binary variable recording the location of preferred instance in a query. No information about the latent function is provided, e.g., its analytic form, convexity, or regularity.

### 2.2 AMORTIZATION

Amortization is a data-driven approach that involves training a model, often a neural network, on a large set of pre-collected tasks. The goal is to enable the model to capture and leverage shared information across these tasks, thus learning how to make predictions efficiently on similar tasks in the future. This process allows the model to perform approximate inference rapidly during the inference stage, since knowledge from related tasks encountered during training has already been internalized.

Recently, amortization has been introduced to BO, either amortizing the surrogate model (Müller et al., 2023; Chang et al., 2024), the acquisition function (Swersky et al., 2020; Volpp et al., 2020), or directly end-to-end training system (Chen et al., 2017; 2022; Yang et al., 2024; Maraval et al., 2023). Our work aims to employ a fully end-to-end trained system that learns how to optimize functions directly from preference data.

Designing a task-specific architecture is crucial to efficiently utilize data in amortized learning. Conditional neural processes (CNPs; Garnelo et al. 2018), as a meta-learning framework built on Deep Sets (Zaheer et al., 2017), are particularly well-suited for tasks like BO (Maraval et al., 2023; Müller et al., 2023), where historical queries are permutation-invariant. Transformer-based neural processes (Nguyen & Grover, 2022; Müller et al., 2022; Chang et al., 2024) extend this idea by leveraging the attention mechanism to more effectively capture interactions between points.

### 2.3 REINFORCEMENT LEARNING

Reinforcement Learning (RL) has recently found success in the Black-Box Optimization realm (Volpp et al., 2020; Hsieh et al., 2021; Maraval et al., 2023), due to its ability to yield *non-myopic* acquisition strategies. This leads to better performances since these strategies consider the influence of future evaluations up to a given horizon determined by the available budget $T$. An RL problem is defined as a Markov Decision Process (MDP) $\mathcal{M} = (\mathcal{S}, \mathcal{A}, \mathcal{P}, \mathcal{R}, \gamma)$, $\mathcal{S}$ and $\mathcal{A}$ stand for the state and action spaces, respectively, $\mathcal{P} : \mathcal{S} \times \mathcal{A} \times \mathcal{S} \to [0, 1]$ is the state transition model, and $\mathcal{R}$ is the reward function which encompasses a given optimization goal. The goal is to learn a probability distribution over state-action pairs $\pi(\boldsymbol{a}_t | \boldsymbol{s}_t)$, referred to as a policy, such that $\pi$ maximizes the discounted expected returns with discount factor $\gamma \in [0, 1]$, which defines the degree of myopia of the policy.

## 3 PREFERENTIAL AMORTIZED BLACK-BOX OPTIMIZATION (PABBO)

To solve the problem introduced in Section 2.1 in an amortized manner, PABBO proceeds as illustrated in Figure 2. In brief, PABBO directly outputs acquisition values for candidate pairs through the policy $\pi_\theta$, which is parameterized by a transformer backbone and an *acquisition head*. To stabilize the training, we additionally employ a *prediction head* for an auxiliary preference prediction task. Section 3.1 describes the MDP we employ for learning the policy $\pi_\theta$, Section 3.2 details the conception of the pre-training dataset accordingly to the neural process framework, and Section 3.3 focuses on the transformer architecture and the loss functions involved in its actual training.

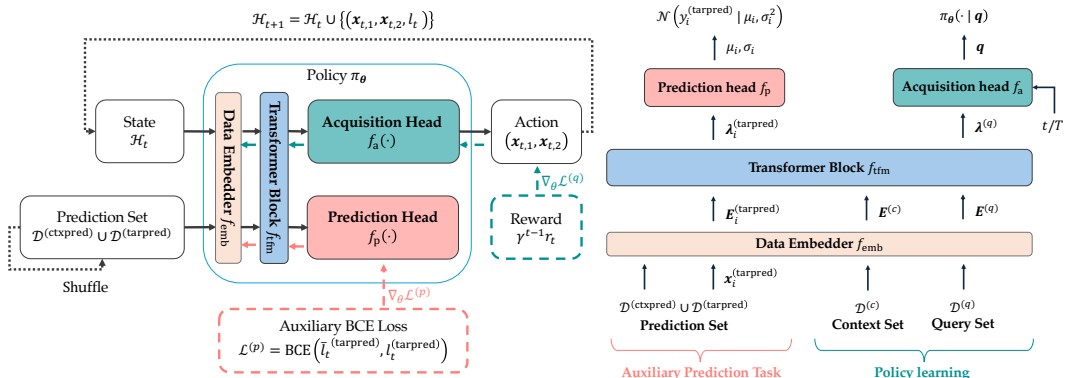

Figure 2: Flowchart of PABBO (left), together with a zoom on our proposed transformer block (right). We use reinforcement learning to guide the acquisition head in proposing valuable query pairs, and apply an auxiliary BCE loss to the prediction head to stabilize the training of the transformer.

### 3.1 POLICY LEARNING

Unlike current PBO methods, we leverage RL to learn a non-myopic policy $\pi_\theta$. For a single task, our RL problem is formulated as follows:

- **States**: $s_t = [\mathcal{H}_t, t, T]$, where $\mathcal{H}_t = \{(\boldsymbol{x}_{i,1}, \boldsymbol{x}_{i,2}, l_i)\}_{i=1}^{t-1}$ contains the optimization history made of $(t-1)$-step queries and preferences.
- **Actions**: $a_t = (\boldsymbol{x}_{t,1}, \boldsymbol{x}_{t,2})$, the next query pair.
- **Rewards**: $r_t = \max_{1 \le i \le t} \max\{y_{i,1}, y_{i,2}\}$, the best objective function values observed so far with $y_{i,1}$ and $y_{i,2}$ being associated to inputs $\boldsymbol{x}_{i,1}$ and $\boldsymbol{x}_{i,2}$, respectively.

State transitions consist in updating the optimization history with the selected next query and preference feedback, $\mathcal{H}_{t+1} = \mathcal{H}_t \cup \{(\boldsymbol{x}_{t,1}, \boldsymbol{x}_{t,2}, l_t)\}$. We seek a policy maximizing the rewards along the entire optimization trajectory while performing well on average across tasks sampled from $p_{\text{tasks}}$. Rewards $r_t$ denote the maximal objective function values observed at step $t$. Such values are always available in synthetic pre-training datasets, like GP draws. If the latent function values are unavailable during pre-training, ranking data can be used to obtain a score for each point as the reward. We show in Section 5.4 that this substitution does not greatly alter performance.

### 3.2 PRE-TRAINING DATA ARCHITECTURE

The PABBO workflow involves a pre-training phase, where a policy is learned on all meta-tasks, and an inference phase, where the trained model is applied to predict acquisition values on a new task directly. Our approach effectively amortizes the optimization process over multiple tasks, allowing rapid inference by leveraging the shared structure learned during pre-training. The pre-training dataset for each meta-task features three parts:

- A **context set** $\mathcal{D}^{(c)} = \{(\boldsymbol{x}_{i,1}^{(c)}, \boldsymbol{x}_{i,2}^{(c)}, l_i^{(c)})\}_{i=1}^{t-1}$ containing all past queries and preferences.
- A **query set** $\mathcal{D}^{(q)} = \{(\boldsymbol{x}_i^{(q)})\}_{i=1}^S$ consisting of $S$ locations that the model considers for the next step, which will be further combined into a candidate query pair set $Q = \{(\boldsymbol{x}_i^{(q)}, \boldsymbol{x}_j^{(q)}) \mid \boldsymbol{x}_i^{(q)}, \boldsymbol{x}_j^{(q)} \in \mathcal{D}^{(q)}, i \ne j\}$ of size $M$. In a training iteration, we randomly sample $S$ query points, and candidate query pairs are chosen from all possible pairwise combinations within this set. At inference time, query points are sampled across the search space from a quasi-random Sobol sequence, and all possible pairs from the full set of combinations are considered. $S$ should be set to a sufficiently large value and scaled with data dimension to ensure adequate exploration.
- A **prediction set** $\mathcal{D}^{(p)} = \{(\boldsymbol{x}_{i,1}^{(p)}, \boldsymbol{x}_{i,2}^{(p)}, l_i^{(p)})\}_{i=1}^N$ containing $N$ queries and preferences. Similarly, we generate $\mathcal{D}^{(p)}$ by sampling pairwise combinations of a set of random points. This prediction set is used to compute an auxiliary binary classification loss: for each optimization step, we randomly sample a subset of $\mathcal{D}^{(p)}$ which refers as *prediction context set* $\mathcal{D}^{(\text{ctxpred})} = \{(\boldsymbol{x}_{i,1}^{(\text{ctxpred})}, \boldsymbol{x}_{i,2}^{(\text{ctxpred})}, l_i^{(\text{ctxpred})})\}_{i=1}^{N^{(\text{ctxpred})}}$, and all the individual points from the

remaining $P = N - N^{(\text{ctxpred})}$ pairs are collected as a *prediction target set* $\mathcal{D}^{(\text{tarpred})} = \{(\boldsymbol{x}_i^{(\text{tarpred})})\}_{i=1}^{N^{(\text{tarpred})}}$. The model is required to estimate the function values for each point in $\mathcal{D}^{(\text{tarpred})}$ conditioned on $\mathcal{D}^{(\text{ctxpred})}$, and further recover the preference relationship of a pair.

It is worth mentioning that no reward information can leak from the query set to the prediction set as entirely different instances are assigned to each set.

### 3.3 MODEL ARCHITECTURE

PBBO differs from standard BO in two aspects: (1) preferential outcomes instead of exact latent function values are observed, and (2) a pair of points is expected as the next query instead of a single point. To address these challenges, PABBO leverages novel data encoding and decoding structures, efficiently handling pairwise preference data. At the center of PABBO's architecture lie conditional neural processes, a class of models quite amenable for handling the sets that naturally arise due to the permutation invariance of the optimization history. Specifically, the PABBO model architecture, presented in Figure 2, consists of four main components: the data embedder $f_{\text{emb}}$, transformer block $f_{\text{tfm}}$, prediction head $f_p$, and the acquisition head $f_a$.

Firstly, inputs from all parts of the dataset are mapped into a shared embedding space via the **data embedder** $f_{\text{emb}}$. The first and second instance within a pair $\boldsymbol{x}_{i,1}, \boldsymbol{x}_{i,2}$ and their corresponding preference $l_i$ are encoded separately with individual MLPs into embeddings of the same size and added up as $\mathbf{E} = \mathbf{E}^{(\boldsymbol{x}_{i,1})} \oplus \mathbf{E}^{(\boldsymbol{x}_{i,2})} \oplus \mathbf{E}^{(l_i)}$. For parts $\mathcal{D}^{(q)}$ and $\mathcal{D}^{(\text{tarpred})}$ containing only a single instance $\boldsymbol{x}_i$, $\mathbf{E} = \mathbf{E}^{(\boldsymbol{x}_i)}$. This gives us embeddings $\mathbf{E}^{(c)}, \mathbf{E}^{(p)}, \mathbf{E}^{(\text{ctxpred})}$ and $\mathbf{E}^{(\text{tarpred})}$ corresponding to $\mathcal{D}^{(c)}, \mathcal{D}^{(q)}, \mathcal{D}^{(\text{ctxpred})}$ and $\mathcal{D}^{(\text{tarpred})}$, respectively. More details and a discussion related to the choice of this particular architecture can be found in Appendix A.

Next, embeddings are passed through the **transformer block** $f_{\text{tfm}}$, which captures the interactions between different parts of a meta-dataset. Dependencies between elements are controlled with masking in self-attention layers of $f_{\text{tfm}}$ (see Figure S1). Precisely, $\mathcal{D}^{(c)}$, the observations collected by the current step, can be attended by other elements in $\mathcal{D}^{(c)}$ and all the query points in $\mathcal{D}^{(q)}$, while each query point can only be attended by itself. Our interest is the Transformer outputs for query set $\boldsymbol{\lambda}^{(q)} = f_{\text{ftm}}(\mathbf{E}^{(q)})$. The same rule applies for the prediction context set $\mathcal{D}^{(\text{ctxpred})}$ and prediction target set $\mathcal{D}^{(\text{tarpred})}$, leading to the transformer outputs $\boldsymbol{\lambda}^{(\text{tarpred})} = f_{\text{ftm}}(\mathbf{E}^{(\text{tarpred})})$. These outputs $\boldsymbol{\lambda} = [\boldsymbol{\lambda}^{(q)}, \boldsymbol{\lambda}^{(\text{tarpred})}]$ are then processed with different decoders according to their specified task.

Subsequently, $\boldsymbol{\lambda}^{(q)}$ are combined into pairs $\{(\boldsymbol{\lambda}_i^{(q)}, \boldsymbol{\lambda}_j^{(q)}) \mid (\boldsymbol{x}_i^{(q)}, \boldsymbol{x}_j^{(q)}) \in Q)\}$ according to the candidate query pair set $Q^*$. Each pair $(\boldsymbol{\lambda}_i^{(q)}, \boldsymbol{\lambda}_j^{(q)})$ passes through the **acquisition head** $f_a$ along with current step $t$ and budget $T$ to predict the acquisition function value $\mathbf{q}_{i,j} = f_a(\boldsymbol{\lambda}_i^{(q)}, \boldsymbol{\lambda}_j^{(q)}, t, T)$ for associated query pair. This enables creating a policy for proposing the next-step query $(\boldsymbol{x}_{t,1}, \boldsymbol{x}_{t,2})$ from $Q$, mapping outputs of $f_a$, $\mathbf{q} = \{\mathbf{q}_{i,j} | (\boldsymbol{x}_i^{(q)}, \boldsymbol{x}_j^{(q)}) \in Q\}$, to a categorical distribution *via* SoftMax:

$$\pi_\theta((\boldsymbol{x}_i^{(q)}, \boldsymbol{x}_j^{(q)}) \mid \mathcal{H}_t, t, T) = \frac{\exp \mathbf{q}_{i,j}}{\sum_{\mathbf{q}_m \in \mathbf{q}} \exp \mathbf{q}_m}. \tag{2}$$

As defined in Section 3.1, the best objective function value obtained so far is chosen as the immediate reward when training this policy, since the goal is to find the global optimum as efficiently as possible. To further pursue a non-myopic solution, we consider the cumulative reward across the entire query trajectory, incorporating a discount factor $\gamma$, which controls the rate at which future rewards are discounted. The loss writes as the negative expected reward over the entire optimization trajectory:

$$\mathcal{L}^{(q)} = -\sum_{t=1}^{T} \gamma^{t-1} r_t \log \pi_\theta((\boldsymbol{x}_{t,1}^{(q)}, \boldsymbol{x}_{t,2}^{(q)}) \mid \mathcal{H}_t, t, T). \tag{3}$$

In addition to the forward process $f_a$, reward sparsity is handled by introducing an auxiliary path leading to the **prediction head** $f_p$, encouraging the learning of latent function shape from preferences. For each prediction target point $\boldsymbol{x}_i^{(\text{tarpred})} \in \mathcal{D}^{(\text{tarpred})}$, $f_p$ parameterizes a Gaussian distribution:

$$p\left(y_i | \boldsymbol{x}_i^{(\text{tarpred})}\right) = \mathcal{N}\left(y_i^{(\text{tarpred})}; (\mu_i, \sigma_i^2) := f_p\left(\boldsymbol{\lambda}_i^{(\text{tarpred})}\right)\right). \tag{4}$$

---

* To reduce computational costs from evaluating all the possible pairwise combinations between query points, which are of size $O(S^2)$, we randomly sample $M \ll S^2$ pairs as $Q$ for each meta-task during pre-training.

Using independent Gaussian likelihoods is common in the Neural Processes community (Garnelo et al., 2018). Correlations can be accounted for if the downstream task requires it (Markou et al., 2022). A similar choice is often found in PBO, using the probit likelihood (Brochu et al., 2010).

At any two target points $\boldsymbol{x}_i^{(\text{tarpred})}$ and $\boldsymbol{x}_j^{(\text{tarpred})}$ that constitute a query pair in $\mathcal{D}^{(p)}$, the predicted latent function values $\bar{y}_i^{(\text{tarpred})}$ and $\bar{y}_j^{(\text{tarpred})}$ are obtained by sampling from $p$, thus capturing the randomness in the process as described by Equation 1. The final preference is $\bar{l}^{(\text{tarpred})} := \sigma(\bar{y}_j^{(\text{tarpred})} - \bar{y}_i^{(\text{tarpred})})$ with $\sigma$ the Sigmoid function. As mentioned, at each optimization step, we generate a prediction task with different contexts by shuffling the prediction set $\mathcal{D}^{(p)}$, creating different splits of $\mathcal{D}^{(\text{ctxpred})}$ and $\mathcal{D}^{(\text{tarpred})}$ to fully utilize the data. $f_p$ is trained by minimizing the binary cross entropy (BCE) loss between predicted preferred location $\bar{l}^{(\text{tarpred})}$ and the ground truth $l^{(\text{tarpred})}$ on all the prediction tasks:

$$\mathcal{L}^{(p)} = -\sum_{t=1}^{T}\sum_{i=1}^{P}\left[ l_{t,i}^{(\text{tarpred})} \log(\bar{l}_{t,i}^{(\text{tarpred})}) + (1 - l_{t,i}^{(\text{tarpred})}) \log(1 - \bar{l}_{t,i}^{(\text{tarpred})}) \right], \qquad (5)$$

where $l_{t,i}^{(\text{tarpred})}$ and $\bar{l}_{t,i}^{(\text{tarpred})}$ represent the ground truth and predicted preference between a possible pair of prediction target points at step $t$, respectively.

The final objective $\mathcal{L}$ for policy learning combines the two losses: $\mathcal{L} := \mathcal{L}^{(q)} + \lambda\mathcal{L}^{(p)}$. We fix $\lambda = 1$ in all the experiments. In practice, we warm up the model with only the prediction task by setting $\mathcal{L} = \mathcal{L}^{(p)}$ in the initial training steps, the rationale being that the policy can only perform well once the shape of the latent function has been learned. Algorithm 1 describes the complete meta-learning training procedure, and Algorithm S1 describes how PABBO operates at test-time on a new task.

---

**Algorithm 1** Preferential Amortized Black-box Optimization (PABBO)

---

**Require:** $K$ meta-tasks, candidate query pair set size $M$, discount factor $\gamma$, query budget $T$
  **for** each training iteration **do**
    initialize $\mathcal{D}^{(p)}, \mathcal{D}^{(q)}$, sample $Q$ of size $M$, set $\mathcal{H}_1 \leftarrow \{\emptyset\}$
    **for** $t = 1, \dots, T$ **do**                                         ▷ start policy learning
      $(\boldsymbol{x}_{t,1}, \boldsymbol{x}_{t,2}) \sim \pi_\theta(\cdot \mid \mathcal{H}_t, t, T)$         ▷ sample next query pair from policy
      $l_t = \boldsymbol{x}_{t,1} \succ \boldsymbol{x}_{t,2}$         ▷ comparison of two sampled query points
      $r_t = \max_{1 \le i \le t} \max\{y_{t,1}, y_{t,2}\}$         ▷ collect immediate reward
      $\mathcal{H}_{t+1} \leftarrow \mathcal{H}_t \cup \{(\boldsymbol{x}_{t,1}, \boldsymbol{x}_{t,2}, l_t)\}$         ▷ update history
      $Q \leftarrow Q \backslash \{(\boldsymbol{x}_{t,1}, \boldsymbol{x}_{t,2})\}$         ▷ update query set
      infer prediction target preference $\bar{l}^{(\text{tarpred})}$         ▷ auxiliary prediction task
      $\mathcal{D}^{(p)} \rightarrow \mathcal{D}^{(\text{ctxpred})} \cup \mathcal{D}^{(\text{tarpred})}$         ▷ update prediction dataset
    **end for**
    Update PABBO using $\mathcal{L} := \mathcal{L}^{(q)} + \lambda\mathcal{L}^{(p)}$ (Equations 3 and 5)
  **end for**

---

## 4   RELATED WORK

**Preferential Black-Box Optimization.** While PBBO has been tackled using techniques like dueling bandits (Bengs et al., 2021) or Reinforcement Learning (Myers et al., 2023; Rafailov et al., 2024), the bulk of the work points to the Preferential Bayesian Optimization field. The latter mostly hinges on an approximate GP surrogate, with vastly different results depending on the approximate inference scheme (Takeno et al., 2023), and several dedicated acquisition functions have been proposed. These include dueling Thompson sampling, or EUBO, a decision-theoretic AF (Lin et al., 2022). Unlike standard BO, deriving regret bounds for PBO is cumbersome. Only one work has done so (Xu et al., 2024), although convergence can be demonstrated, e.g. for dueling Thompson sampling (Astudillo et al., 2024). Lastly, Bayesian Neural Networks were also trialed (Huang et al., 2022).

PBO has been extended to various settings: handling a batch of queries $q > 2$ (Siivola et al., 2021; Astudillo et al., 2023), heteroscedastic noise (Sinaga et al., 2024) and multi-objective optimization (Astudillo et al., 2024). On another note, preferential relations $\boldsymbol{x}_1 \succ \boldsymbol{x}_2$ have been leveraged to enhance vanilla BO loops: Hvarfner et al. (2024) investigate a user-defined prior that is integrated to the GP surrogate, whereas Adachi et al. (2024) allow the prior to be iteratively updated.

**Amortization and Meta-Learning.** Our architecture is closely related to Conditional Neural Processes (CNPs) (Garnelo et al., 2018; Kim et al., 2019; Jha et al., 2022; Huang et al., 2023), and its more recent variant, Transformer-based Neural Processes (Nguyen & Grover, 2022; Müller et al., 2022; Chang et al., 2024). CNPs have primarily focused on amortizing the predictive posterior distribution via supervised learning on a collection of datasets, whilst our approach employs reinforcement learning to train an optimization policy.

In recent years, multiple works have explored training neural networks to directly amortize black-box optimization (Chen et al., 2017; Yang et al., 2024; Maraval et al., 2023; Amos, 2023; Song et al., 2024; Chen et al., 2022). These methods typically rely on direct function utilities for training. Recent advancements also include the use of large language models to assist in Bayesian Optimization (Liu et al., 2024; Chen et al., 2024; Aglietti et al., 2024). However, to the best of our knowledge, no existing work has yet attempted to directly amortize optimization based on preferential feedback data.

Besides, since PBBO can be viewed as a sequential experimental design (SED) problem (Rainforth et al., 2024), our work is also related to research on amortized SED. This line of work primarily revolves around Deep Adaptive Design (DAD; Foster et al. 2021), which develops a comprehensive framework to amortize experimental design. Subsequent work has extended this approach to settings with implicit likelihoods (Ivanova et al., 2021) and has leveraged reinforcement learning to further improve performance (Blau et al., 2022; Lim et al., 2022). Our work can be seen as a specialized application of SED with specific downstream objectives (Huang et al., 2024).

**Preference tuning for language models.** Finally, recent advancements in preference learning have further expanded its application in tuning the language models, particularly in tasks such as reinforcement learning with human feedback (RLHF) to align large-scale models with user preferences. Christiano et al. (2017) proposes a framework for guiding policy optimization in RL with human preference feedback, which has been applied to several natural language tasks (Zaheer et al., 2017; Stiennon et al., 2020). While those works focus on relatively small models and specific tasks, InstructGPT (Ouyang et al., 2022) extends it to large-scale models like GPT-3, allowing alignment with user instructions across a wide range of tasks. Recently, Rafailov et al. (2024) proposes Direct Preference Optimization (DPO), which shows that language models can be directly trained on preference data without explicit reward modeling or RL optimization, significantly simplifying the preference learning pipeline. While PABBO and these works both leverage preference feedback, they serve fundamentally different purposes: language model alignment aims to steer model behavior towards desired outputs in text spaces, whereas PABBO focuses on efficient optimization of black-box functions, requiring different architectural designs and optimization strategies.

## 5 Experiments

We evaluate PABBO on synthetic functions, described in Section 5.1, and against real-world examples from the BO literature: hyperparameter optimization (Section 5.2) and human preferences datasets (Section 5.3). Subsequently, Section 5.4 presents a series of ablation studies: replacing the latent values in the rewards by pure rankings, examining different discount factor values $\gamma$ in Equation 3, and varying the number of query locations on which the acquisition function values are predicted.

**Baselines.** Our approach is benchmarked against five GP-based acquisition strategies from the PBO realm: qEUBO (Astudillo et al., 2023), qEI / qNEI (Siivola et al., 2021), qTS (Hernández-Lobato et al., 2017) and MPES (Nguyen et al., 2020). We also include a strategy where candidate pairs are acquired at random. While qEUBO is a principled preferential acquisition strategy, qEI and qTS are effectively batch BO strategies used in a PBO setting, a solution commonly employed in the field. For instance, qTS samples two draws from the posterior, maximizes each draw, and provides each maximizer as a candidate pair for comparison. Further details can be found in Appendix D.3.

As performance metric, we consider the simple regret $f(\boldsymbol{x}^*) - f(\boldsymbol{x}_T^{\text{best}})$ where $\boldsymbol{x}_T^{\text{best}} = \arg\max_{t \in \{1,\dots,T\}} f(\boldsymbol{x}_t)$. For pre-training, the (discounted) cumulative simple regret was used (Equation 3), sending a stronger and less sparse signal than simple regret. But because our goal is human preferences optimization, e.g., finding the *best* sushi product for a given user, we report simple regret at inference time. However, for completeness, Figure S2 reports cumulative simple regret.

**Implementation.** PABBO is implemented using `PyTorch` (Paszke et al., 2019). Hyperparameter settings can be found in Appendix D.1. For qEUBO, qEI, qNEI, we used the implementation from

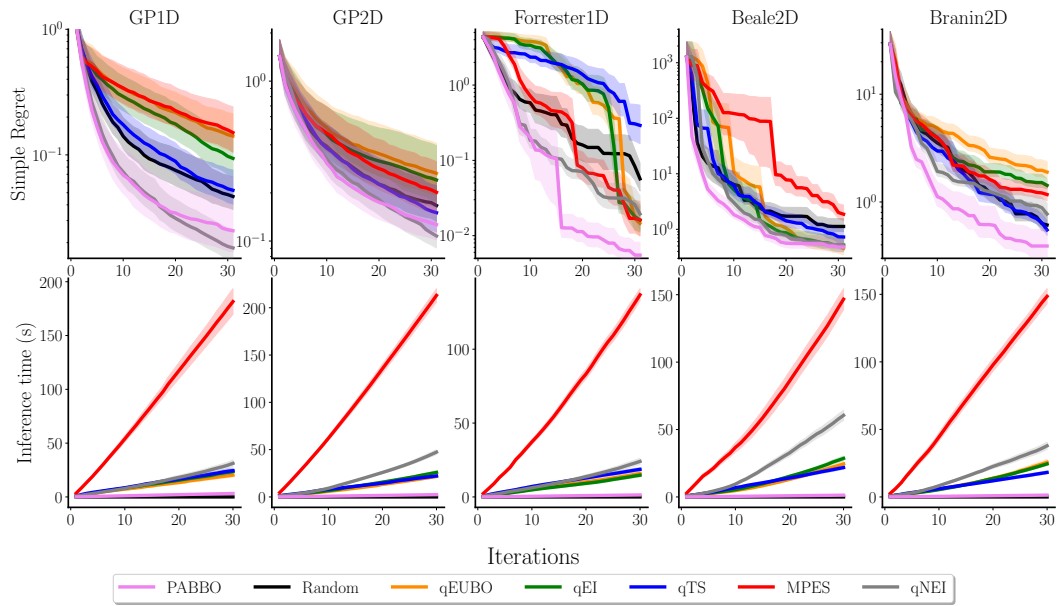

Figure 3: Simple regret and inference time on synthetic examples. Mean with 95% confidence intervals computed across 30 runs with random starting pairs. **PABBO consistently achieved the lowest simple regret across all tasks, except for GP cases where it performs comparably to qNEI, while offering a 10× speedup.**

the `BoTorch` library (Balandat et al., 2020). For qTS and MPES our implementation is described in Appendix D.3. The code to reproduce our experiments is available at https://github.com/xinyuzc/PABBO.

## 5.1 SYNTHETIC FUNCTIONS

We begin by benchmarking PABBO on two types of synthetic functions. The first type involves 30 draws from the same GPs that were used to build the pre-training dataset (See Appendix C for the data generation process). We refer to this case as the *in-distribution*. The second type refers to classical test functions: the Forrester, Branin, and Beale functions. These functions were not part of the pre-training dataset used by PABBO, hence we refer to this case as *out-of-distribution*.

At each training epoch, we generate $B$ different GP samples, to which a quadratic bowl is added to increase the chances of a draw exhibiting a global optimum. Then, we draw $200 \times D$ points, where $D$ is the dimension of $\boldsymbol{x}$, half of which for the auxiliary prediction task and the rest for the optimization task. For synthetic functions, the query set size was set to $S = 256$.

PABBO consistently ranks either as best or second to best method in terms of simple regret (Figure 3). While this is expected for the *in-distribution* case, given its low dimensionality compared to our model architecture, in the *out-of-distribution* setting, PABBO also achieves the lowest simple regret across all three examples. Of note, the random strategy often outperforms certain GP baselines. Regarding inference speed, PABBO offers an average 12-fold speed-up over the fastest GP-based strategy across 5 problems, consistently over the whole trial.

Additionally, PABBO successfully recovers at least one of the three global optima of the Branin function (Figure S11, left panel), and correctly identifies the global optimum of the Forrester function (right panel). Beyond convergence to an optimum, PABBO also accurately learned the shape of each function from preference data, a modality that only allows identification up to a monotonic transformation.

## 5.2 HYPERPARAMETER OPTIMIZATION

Next, we consider hyperparameter optimization (Pineda Arango et al., 2021). This challenge emulates the task an ML expert would face when optimizing the hyperparameter of a given model. In real-

Figure 4: Simple regret on different search spaces from the HPO-B benchmark and human preferences datasets. Mean with 95% confidence intervals computed across 30 runs with random starting pairs. Attached to each PABBO baseline is a number corresponding to $S$, the size of the query set. **On average, PABBO ranks first on HPO-B tasks and second for human preferences datasets.**

world scenarios, experts often find it easier to express preferences between pairs of hyperparameter configurations, rather than assigning precise scalar utilities. The HPO-B benchmark contains multiple search spaces, each corresponding to the optimization of different ML models.

Each search space contains multiple meta-datasets from different meta-tasks, and each meta-dataset collects a list of hyperparameter configurations with their response values, $\{(\boldsymbol{x}_i, y_i)\}_{i=1}^n$, allowing us to simulate preference for any two configurations given their responses. All the meta-datasets within a search space have been categorized into three splits: meta-train, meta-validation, and meta-test.

We choose three representative search spaces and pre-train our model on the meta-train split associated with each space, further splitting each set into two equal-sized, non-overlapping parts beforehand for the auxiliary prediction and optimization tasks (see Appendix D.2). For evaluation, we take all points from each meta-dataset of the meta-test split of the search space, allowing us to assess the average performance of our model and the baselines across all meta-datasets.

Overall, PABBO achieves the best performance, closely followed by the qNEI baseline (Figure 4, first three panels). It is worth noticing that on these higher-dimensional problems, all methods now clearly outperform the random acquisition strategy.

## 5.3 HUMAN PREFERENCES

We experiment with our model pre-trained with synthetic GP samples on two real-world human preference datasets: the Candy* and Sushi* datasets. The Candy dataset provides the full ranking of 85 different candies from pairwise preference via online studies. Each candy object contains 2 continuous features, `sugarpercent` and `pricepercent`. The Sushi dataset collects a preference score by asking users to rate 100 different types of sushi on a five-point-scale. The overall score for each sushi is obtained by averaging across users. Each sushi contains 4 continuous features. Following Siivola et al. (2021), we generalize both datasets to continuous space by performing linear extrapolation, and the out-of-bound values are filled with their nearest neighbors in the original dataset.

For both examples, we report the simple regret achieved by our model when using a query set of size $S = 512$. (Figure 4, fourth and fifth panels). On these challenging *out-of-distribution* preference learning tasks, PABBO remains competitive with qNEI and MPES, achieving the second and third place for the candy and sushi datasets, respectively. Both problems are characterized by a large variance in the results for all baselines.

## 5.4 ABLATION STUDY

This last experimental section investigates whether certain modeling choices and hyperparameter values may have a significant impact on the performance of PABBO. A set of more detailed ablation studies can be found in Appendix B.3.

**Access to latent values during pre-training.** Even though latent values are always accessible for synthetic datasets used for pre-training, this might not be the case for real-world preferential meta datasets. A simple solution for proposing rewards without access to the true utility is to use rankings from a fully ranked set to approximate the latent function values. We trained a 1-and-2-dimensional models using a pre-training dataset of GP samples, substituting classical rewards for each meta-task

---

* Available at data/candy-power-ranking    * Available at https://www.kamishima.net/sushi/

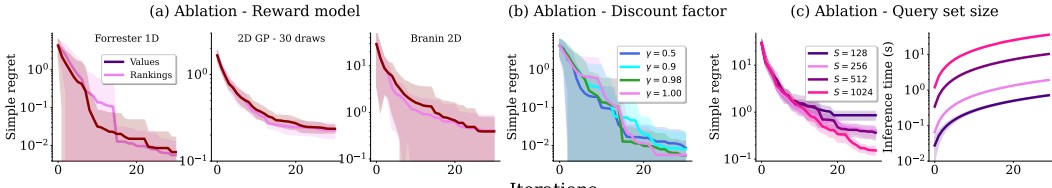

Figure 5: Ablation studies. Mean with 95% confidence intervals computed across 30 runs with random starting pairs. **On low dimensional examples, PABBO maintains similar performance when substituting latent function values for rankings, and performs as expected when varying the discount factor $\gamma$ and query set size $S$.**

by the rankings of corresponding samples. On the 1-dimensional Forrester function, this alternative version of PABBO still yields decent performances (Figure 5(a)).

**Discount factor $\gamma$.** PABBO learns a policy $\pi_\theta$ by taking the future discounted rewards into account across a time horizon $T$. A smaller discount factor prioritizes immediate queries, giving more weight to the early steps in the optimization process. In contrast, as $\gamma$ approaches 1, the future queries become increasingly important, reflecting a more balanced consideration of both immediate and future rewards. We test with different discount factors on the 1-dimensional Forrester problem. The results in Figure 5(b) align with our expectation. When $\gamma$ is small, we observe faster convergence in the early stages. However, since the future rewards are not sufficiently considered, the average performance over larger steps is less favorable compared to using a larger $\gamma$, which balances both immediate and future rewards, resulting in better long-term optimization outcomes.

**Query set size $S$.** At inference time, a continuous search space is discretized into $S$ query locations sampled from a quasi-random Sobol sequence, used to build a set of candidate query pairs consisting of all possible pairwise combinations, resulting in $O(S^2)$ query pairs. $S$ introduces a trade-off between performance and inference time: a denser grid guarantees adequate exploration of the search space, but longer inference times. For the 2-dimensional Branin problem, increasing the grid size $S \in [128, 256, 512, 1024]$ leads to increased inference time, a consequence of the need to embed an increasing number of candidate pairs (Figure 5(c)). In terms of simple regret, a grid of size $S = 1024$ achieves the best result, whereas $S = 256$ offers the best speed/accuracy tradeoff.

## 6 DISCUSSION

We introduced PABBO, a method to conduct preferential black-box optimization in a fully amortized manner. Leveraging an offline training phase on both synthetic and existing datasets, PABBO achieves considerably reduced inference time jointly with top-ranking performances compared to GP alternatives, based on our benchmarks. One way to explain this gap is that our pre-training dataset contains latent function values, which are always accessible for synthetic datasets, thus giving PABBO a significant advantage over preferential GPs. This being said, a dedicated ablation study demonstrated that replacing latent function values with pure rankings did not significantly affect PABBO's results.

Due to the model architecture and the difficulty of the task, learning to optimize from preferential data, PABBO requires large amounts of pre-collected data. While synthetic datasets might be expressive enough, specific applications falling *out-of-distribution* might pose an issue. More generally, this raises the question of how the pre-training dataset should be constructed. A current limitation of our approach is the size of the query set, $S$, which scales with task dimensionality, increasing inference times (Figure 5). Preliminary results suggest parallelization may help mitigate this issue. Additionally, a promising solution for high-dimensional tasks is developing a dimension-agnostic architecture that can handle inputs of varying dimensions. Currently, PABBO can only handle tasks with the same dimensionality, requiring separate models for tasks with different dimensions. Such a strategy could leverage low-dimensional tasks for high-dimensional ones. This is typically useful if the objective possesses a hidden low-dimensional structure, or decomposes additively. Finally, PABBO directly outputs acquisition values for a candidate pair, modeling the user's latent goal, but does not account for the fact that ultimately, a human is performing the comparison. Introducing dedicated user models like the one presented by Sinaga et al. (2024) could be an interesting extension to this work.

## ACKNOWLEDGEMENTS

XZ, DH and SK were supported by the Research Council of Finland (Flagship programme: Finnish Center for Artificial Intelligence FCAI and decision 341763). SK was also supported by the UKRI Turing AI World-Leading Researcher Fellowship, [EP/W002973/1]. The authors wish to thank Aalto Science-IT project, for the computational and data storage resources provided.

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

# Appendix

The appendix is organized as follows:

- Section A provides details about the inner mechanisms of PABBO. It contains an algorithm for test time inference, an illustrative figure for the masks employed in self-attention layers, a discussion on different modeling choices for PABBO's architecture, and a description of our Batch PABBO extension for high-dimensional scalability.
- Section B contains additional experiments and analyses:
  - Section B.1 presents cumulative regret results across all experiments from the main text.
  - Section B.2 evaluates PABBO's performance on additional high-dimensional test functions, including 6D Ackley, 6D Hartmann, and a 16D HPO-B task.
  - Section B.3 provides comprehensive ablation studies on pre-training dataset composition, auxiliary loss weight, discount factor, and query budget effects.
  - Section B.4 shows a visualization demonstrates about PABBO's capabilities.
- Section C gives the procedure for generating the synthetic datasets used by PABBO during meta-training.
- Section D provides additional details regarding the experiment section. Specifically, a table for PABBO's hyperparameters is provided, along with a description of the HPO-B datasets, the baselines employed, and the hardware used for training and inference.

## A   FURTHER DETAILS ON PABBO

### A.1   TEST-TIME ALGORITHM

---

**Algorithm S1** PABBO - test time algorithm

---

**Require:** Query set size $S$, query budget $T$, initial observations $\mathcal{H}_1$ (possibly empty)

Initialize $\mathcal{D}^{(q)}$ with $S$ quasi-random samples

$Q = \{(\boldsymbol{x}_i, \boldsymbol{x}_j) \,\forall (i,j) \in \{1, \ldots, S\}^2, i \neq j\}$

**for** $t = 1, \ldots, T$ **do**

    $\mathcal{D}^{(c)} \leftarrow \mathcal{H}_t$                                           ▷ update context set

    $\mathbf{E}^{(c)} = f_{\text{emb}}(\mathcal{D}^{(c)}), \mathbf{E}^{(q)} = f_{\text{emb}}(\mathcal{D}^{(q)})$           ▷ embed context set and query set

    $\lambda^{(q)} = f_{\text{tfm}}(\mathbf{E}^{(q)}; \mathbf{E}^{(c)})$       ▷ Transformers output for query set conditioning on context set

    $\mathbf{q} = \{f_a(\lambda_i^{(q)}, \lambda_j^{(q)}, t, T) \,\forall (\boldsymbol{x}_i, \boldsymbol{x}_j) \in Q\}$    ▷ predict acquisition function values for all query pairs

    $\pi_\theta(Q \mid \mathcal{H}_t, t, T) = \text{Categorical}(\mathbf{q})$          ▷ form policy as a Categorical distribution

    $(\boldsymbol{x}_{t,1}, \boldsymbol{x}_{t,2}) \sim \pi_\theta(Q \mid \mathcal{H}_t, t, T)$          ▷ sample next query pair from policy

    $l_t = \boldsymbol{x}_{t,1} \succ \boldsymbol{x}_{t,2}$                          ▷ observe binary preference

    $\mathcal{H}_{t+1} \leftarrow \mathcal{H}_t \cup \{(\boldsymbol{x}_{t,1}, \boldsymbol{x}_{t,2}, l_t)\}$            ▷ update history

    $Q \leftarrow Q \backslash \{(\boldsymbol{x}_{t,1}, \boldsymbol{x}_{t,2})\}$                     ▷ update query set

**end for**

---

### A.2   ARCHITECTURE DETAILS

PABBO employs *separate* MLPs for $\boldsymbol{x}_{i,1}, \boldsymbol{x}_{i,2}$ and $l_i$, mapping each of them to *separate* embeddings of Transformer input dimension as $\mathbf{E}^{(\boldsymbol{x}_{i,1})}, \mathbf{E}^{(\boldsymbol{x}_{i,2})}$, and $\mathbf{E}^{(l_i)}$, with $\oplus$ denoting the component-wise addition. When working on the context set $\mathcal{D}^{(c)}$ and prediction context set $\mathcal{D}^{(\text{ctxpred})}$, which consists of duels $(\boldsymbol{x}_{i,1}, \boldsymbol{x}_{i,2}, l_i)$, the token embedding is obtained by adding the individual embeddings up, i.e. $\mathbf{E} = \mathbf{E}^{(\boldsymbol{x}_{i,1})} \oplus \mathbf{E}^{(\boldsymbol{x}_{i,2})} \oplus \mathbf{E}^{(l_i)}$. When processing a single design from the query set $\mathcal{D}^{(q)}$ or prediction target set $\mathcal{D}^{(\text{tarpred})}$, we directly use the individual embedding for that design as the token

embedding, i.e., $\mathbf{E} = \mathbf{E}^{(\boldsymbol{x}_{i,1})}$. Because $(\boldsymbol{x}_{i,1}, \boldsymbol{x}_{i,2}, l_i)$ forms a single atomic element in our context set, Transformer architecture requires all elements in its input sequence to live in the same embedding space: to pass these preference observations to the transformer, we need to make sure their dimensions are aligned.

Given that PABBO encodes elements separately, for the same type of elements, like $\boldsymbol{x}$, a shared embedder can be used across all three sets. This enables parameter sharing for the embedders, as the same embedder processes all $\boldsymbol{x}$-type input, regardless of which set they belong to. Model complexity benefits from this choice as the need to train separate embedding functions for each set is avoided. Besides, separate embeddings confer distinct semantic meanings to features $\boldsymbol{x}$ and preference labels $l$. Thus, separate MLPs can specialize in their respective tasks: feature embedders can focus on capturing spatial relationships, and preference embedders can focus on binary relationship encoding. This leads to an increased representation power.

On a similar note, one could have settled for using the same embedding for $\boldsymbol{x}_{i,1}$ and $\boldsymbol{x}_{i,2}$. However, in the particular case of preferential feedback, upon summing $\mathbf{E}^{(\boldsymbol{x}_{i,1})}$ and $\mathbf{E}^{(\boldsymbol{x}_{i,2})}$ the order information would have been lost: the network would not know whether $\boldsymbol{x}_{i,1} > \boldsymbol{x}_{i,2}$, or the opposite. Therefore, we opted for separate embedders for $\mathbf{E}^{(\boldsymbol{x}_{i,1})}$ and $\mathbf{E}^{(\boldsymbol{x}_{i,2})}$.

The interactions between different elements in our transformer architecture are controlled through attention masking, which determines which elements can attend to each other. Figure S1 illustrates examples of these masks in self-attention layers. These carefully designed masks ensure that our model processes information in a manner that preserves the causal structure of the preferential optimization problem.

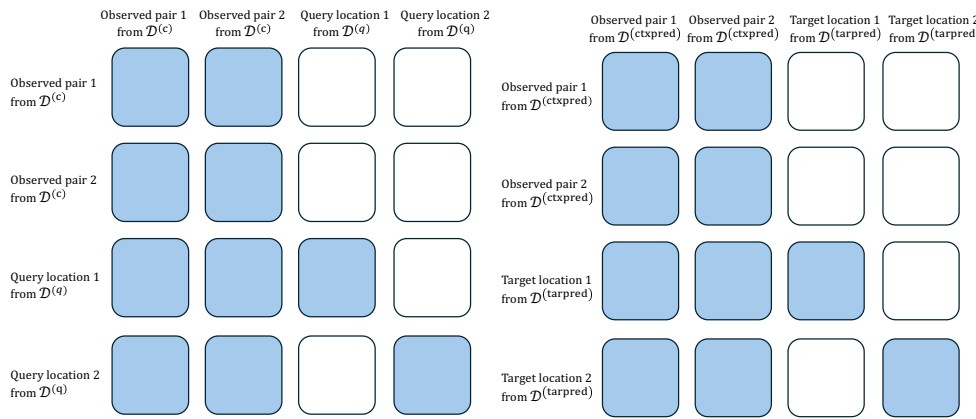

Figure S1: Examples of masks in self-attention layers. Colored squares represent the elements from the left column can attend to the elements on the top in the self-attention layers of the Transformer block $f_{\text{ftm}}$. Left: An example of mask for the acquisition process, involving 2 samples from $\mathcal{D}^{(c)}$ and 2 points from $\mathcal{D}^{(q)}$. Right: Mask example for the auxiliary prediction process, with 2 samples from $\mathcal{D}^{(\text{ctxpred})}$ and 2 points from $\mathcal{D}^{(\text{tarpred})}$.

## A.3 BATCH PABBO FOR HIGH-DIMENSIONAL SCALABILITY

To address scalability challenges in high-dimensional spaces, we developed Batch PABBO, a parallelized extension of our standard approach. While the original PABBO processes a single query set of size $S$, Batch PABBO evaluates $B$ separate query sets of size $S$ simultaneously, selecting the best candidate pair from the combined pool of $B \times S$ candidates: at each optimization step, $B$ candidate pairs are independently proposed from the query sets, and the one with the highest predicted acquisition function value is selected as the next query. This approach effectively increases the exploration coverage while maintaining manageable computational requirements. We show the results of batch PABBO in our additional high-dimensional experiments (Appendix B.2).

## B ADDITIONAL EXPERIMENTS

### B.1 CUMULATIVE REGRET RESULTS

While our main evaluation focuses on simple regret since our primary applications (such as human preference optimization) prioritize finding the optimal solution rather than the path taken, we also analyze the cumulative regret performance of PABBO. This metric is particularly relevant during training, where we use the (discounted) cumulative simple regret as a reward function to provide stronger learning signals (Equation 3). Figure S2 shows the cumulative simple regret across all experiments presented in the main text.

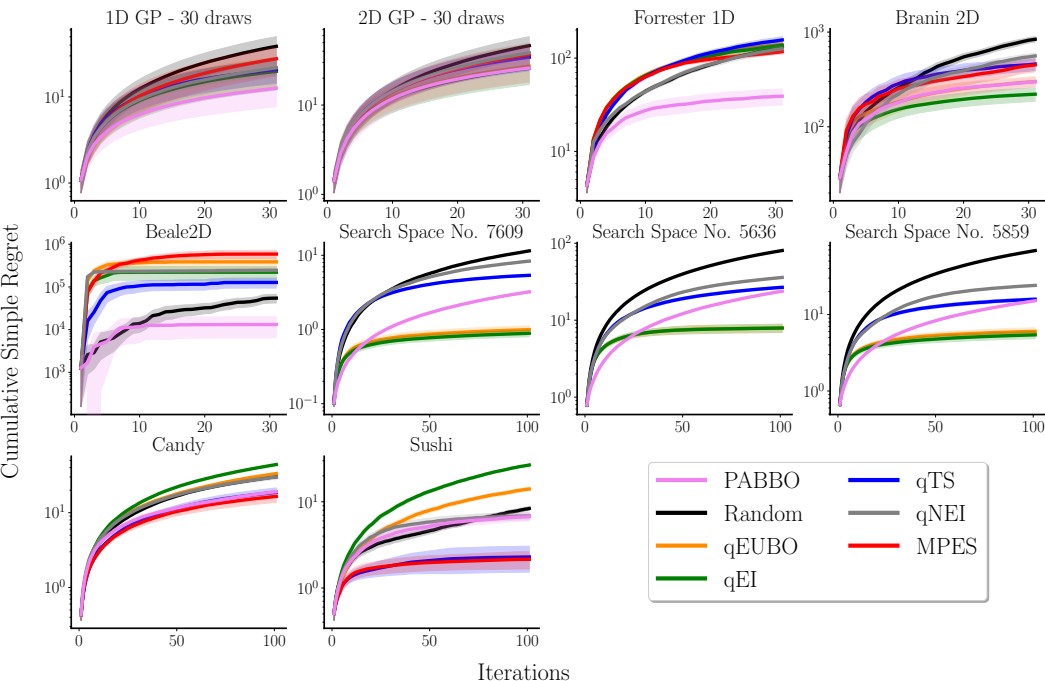

Figure S2: Cumulative simple regret plots for all experiments carried out in the main text. Mean with 95% confidence intervals computed across 30 runs with random starting pairs. **PABBO achieves competitive cumulative regret, ranking second on average across all tasks.**

### B.2 HIGH-DIMENSIONAL FUNCTION EVALUATION

To further evaluate PABBO's performance in higher-dimensional spaces, we conducted additional experiments on three challenging benchmarks: the 6-dimensional Ackley function, the 6-dimensional Hartmann function, and a 16-dimensional search space from the HPO-B benchmark. On the Ackley function (Figure S3), we also tested our Batch PABBO (Appendix A.3) which allows PABBO to evaluate multiple query sets simultaneously. The experiments reveal varied performance across different functions. While PABBO performs well on the Ackley function (Figure S3) and the 16-dimensional HPO-B task (Figure S5), it shows limitations on the Hartmann function (Figure S4). This performance gap on Hartmann likely stems from its multimodal landscape with many local optima, which differs significantly from the functions in our pre-training dataset. This observation suggests that incorporating more diverse function types during pre-training could improve performance on such challenging landscapes.

### B.3 FURTHER ABLATION STUDIES

To provide deeper insights into PABBO's performance characteristics and design choices, we conduct a series of additional ablation studies exploring various aspects of our model.

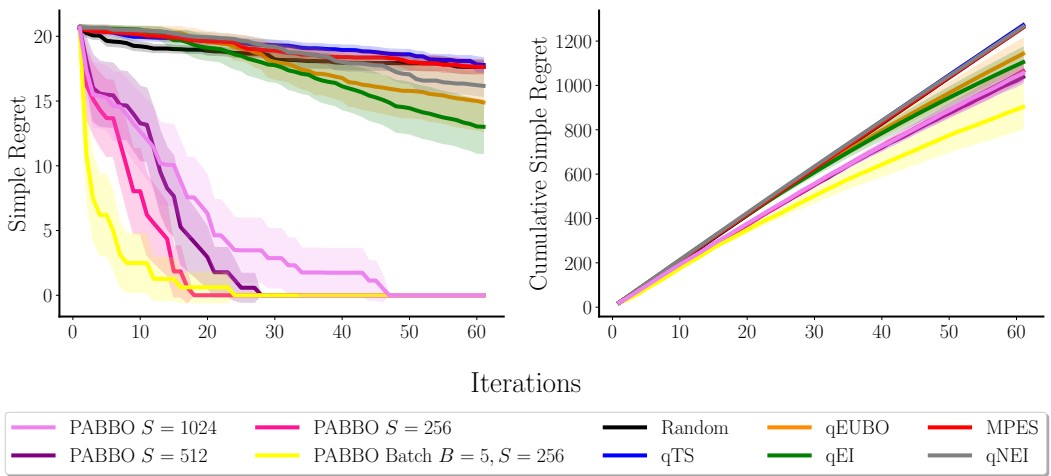

Figure S3: Simple regret and cumulative simple regret on the 6-dimensional Ackley function. Mean with 95% confidence intervals computed across 30 runs with random starting pairs. **Using parallelization, we can mimic a larger query set size for PABBO, thus achieving faster convergence than non-batch PABBO versions, for a significantly faster cumulative inference time (13s compared to 40s for PABBO S = 1024).**

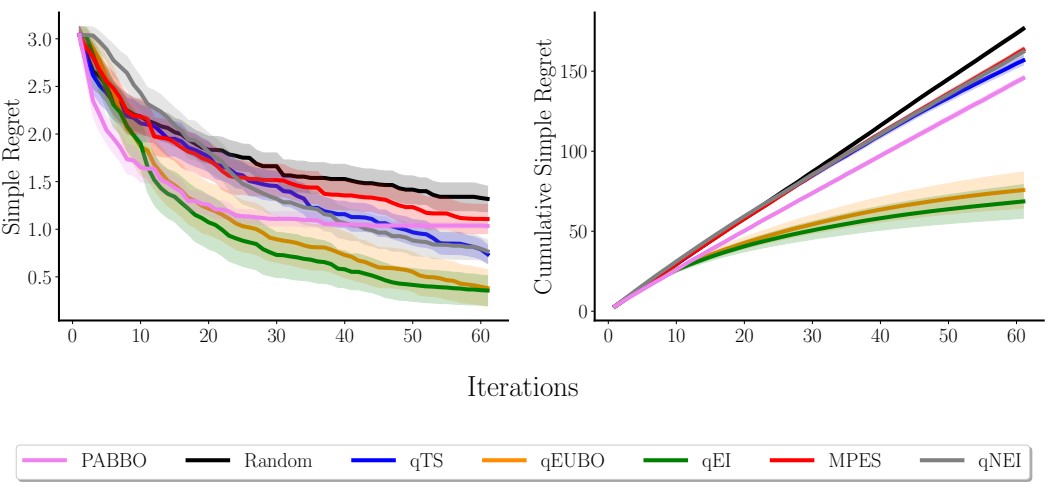

Figure S4: Simple regret and cumulative simple regret on the 6-dimensional Hartmann function. Mean with 95% confidence intervals computed across 30 runs with random starting pairs. We used a query set size $S = 1024$ for PABBO. **PABBO does not match the performance of most GP-based baselines on this example.**

### B.3.1 PRE-TRAINING DATASET COMPOSITION

The composition of the pre-training dataset is critical for the generalization capabilities of our model. To quantify this effect, we compare two variants: PABBO RBF, pre-trained exclusively on GP draws sampled using an RBF kernel, and the standard PABBO, pre-trained on a diverse mixture of kernels as described in Appendix C.

As shown in Figure S6, training on a mixture of kernels leads to better performance compared to using only the RBF kernel. This confirms our hypothesis that a more diverse pre-training dataset enhances the model's ability to generalize to various function landscapes.

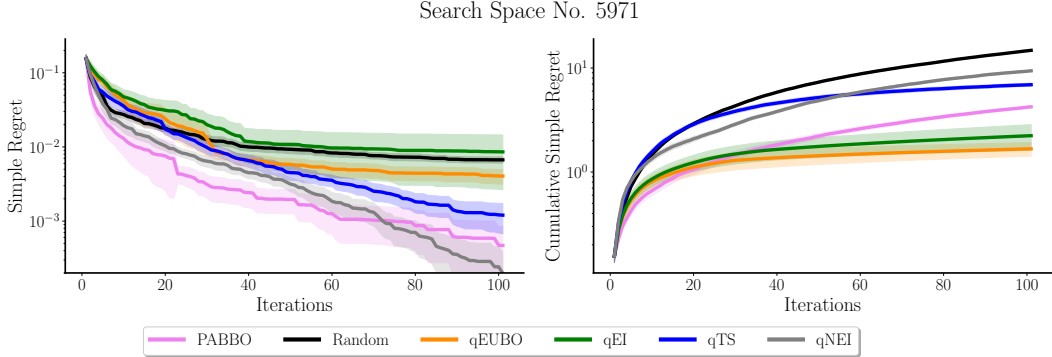

Figure S5: Simple regret and cumulative simple regret on a 16-dimensional search space from the HPO-B benchmark. Mean with 95% confidence intervals computed across 30 runs with random starting pairs. We used a query set size $S = 1024$ for PABBO. **PABBO ranks 2nd in terms of simple regret and 3rd in terms of cumulative simple regret.**

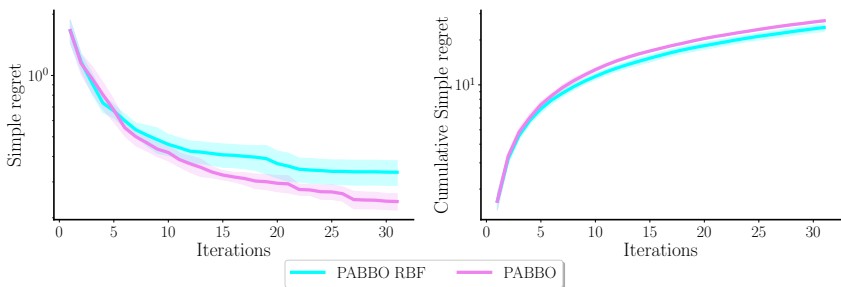

Figure S6: Ablation study on the composition of the pre-training synthetic dataset on 30 2D GP draws, with a query set size $S = 256$. Mean with 95% confidence intervals computed across 5 runs with random starting pairs. PABBO RBF is pre-trained on GP draws sampled using an RBF kernel, whereas PABBO uses the procedure described in Section C. Both baselines use lengthscales $l \sim \mathcal{U}([1.5, 2])$. **As expected, pre-training on a dataset coming from a mixture of kernels rather than only a RBF kernel yields better results, due to a more pronounced diversity of the synthetic samples.**

### B.3.2 LOSS WEIGHT AND EFFECT OF WARM-UP PHASE

We explore the impact of the loss weight hyperparameter $\lambda$ in the combined loss function, as well as the effect of the warm-up phase where the model is initially trained using only the prediction task.

The results in Figure S7 reveal that the warm-up phase plays a crucial role in PABBO's performance. This confirms our intuition that allowing the model to first learn the shape of the latent function stabilizes the subsequent policy learning process. Interestingly, once the warm-up phase is included, we notice that the specific value of $\lambda$ has only a mild effect on the overall performance.

### B.3.3 DISCOUNT FACTOR IN REINFORCEMENT LEARNING

We conduct a more comprehensive ablation study by considering a denser grid of $\gamma$ values, spaced between 0 to 1, with results shown in Figure S8. The conclusion aligns with the main text: smaller $\gamma$ values prioritize immediate rewards, leading to slightly faster convergence in the early stage but worse performance in the long term. Additionally, we observe that $\gamma < 0.5$ deteriorates performance, it might be that sparse reward signals hinder the model convergence. Based on these findings, we recommend using a $\gamma$ close to 1 when the primary goal is minimizing final regret. For scenarios where early-stage performance is crucial, a $\gamma$ between 0.5 and 1.0 can be considered.

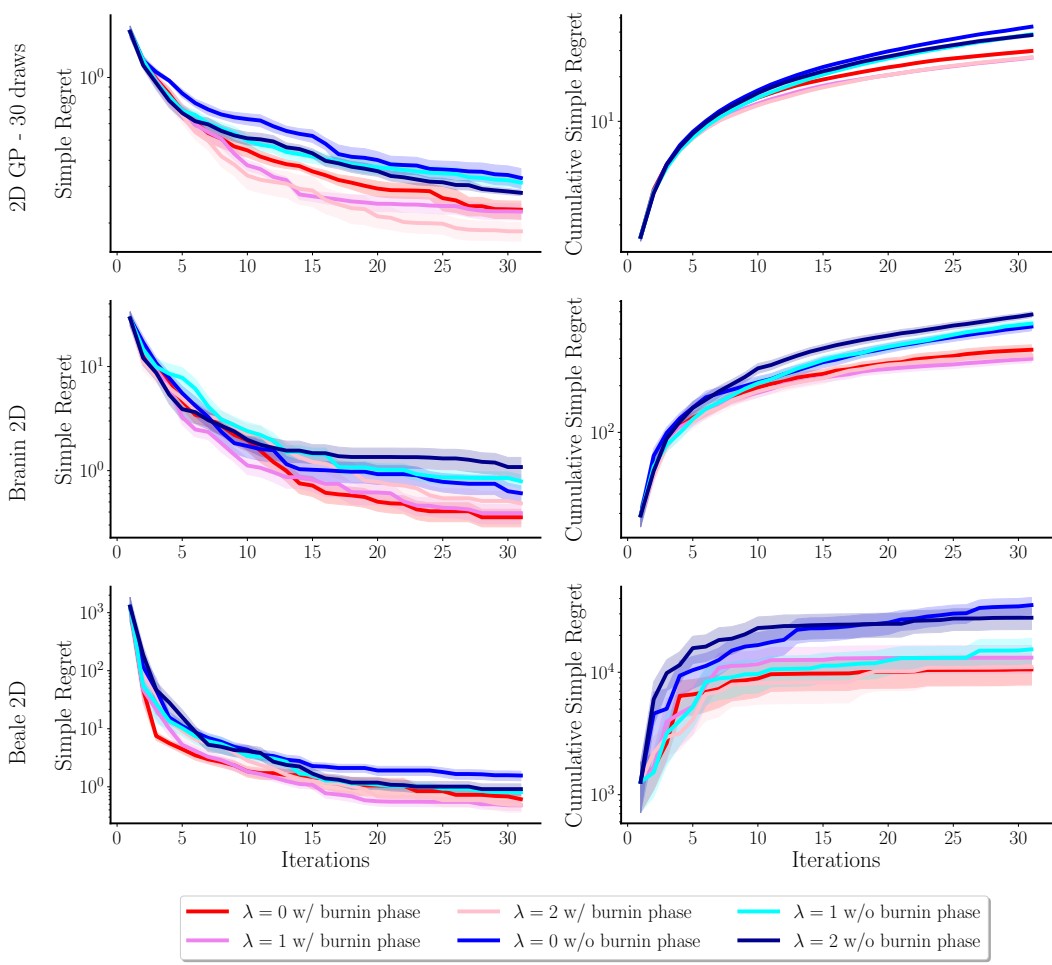

Figure S7: Ablation study on the loss weight hyperparameter $\lambda$ in $\mathcal{L} = \mathcal{L}^{(q)} + \lambda\mathcal{L}^{(p)}$ (Equations 3 and 5) on three 2D functions, with a query set size $S = 256$. During the warm-up phase, PABBO is trained using only the prediction task $\mathcal{L}^{(p)}$. **We observe a stark contrast depending on the presence or not of the warm-up phase, which validates the usefulness of the prediction task, mainly as a tool to stabilize training. Results are mildly affected by different values of $\lambda$.**

### B.3.4 QUERY BUDGET AND TEMPORAL INFORMATION

We now explore how the query budget $T$ used during pre-training affects performance, as well as the impact of passing temporal information $t/T$ to the acquisition head. The results are shown in Figure S9. Our results indicate that including $t/T$ as input to the acquisition head improves the performance, which means the temporal information $t/T$ helps the model better capture the progression of the optimization process, leading to more informed acquisition decisions. Additionally, we observe that when $t/T$ is included, a larger query budget $T$ is associated with faster convergence and the lowest cumulative simple regret.

### B.3.5 ABLATION ON THE PERCENTAGE OF CONTEXT AND TARGET PREDICTION SET

In the main experiments, we set a total number of samples for the prediction set and *randomly* sample $D^{(\text{ctxpred})}$ and assign the remaining pairs as $D^{(\text{tarpred})}$. To examine the effect of the percentage of context and target prediction set, we train different PABBO for prediction task, when setting $N^{(\text{ctxpred})} = 10$ with varying $N^{(\text{tarpred})} = [1, 10, 100]$. The result is shown in Figure S10.

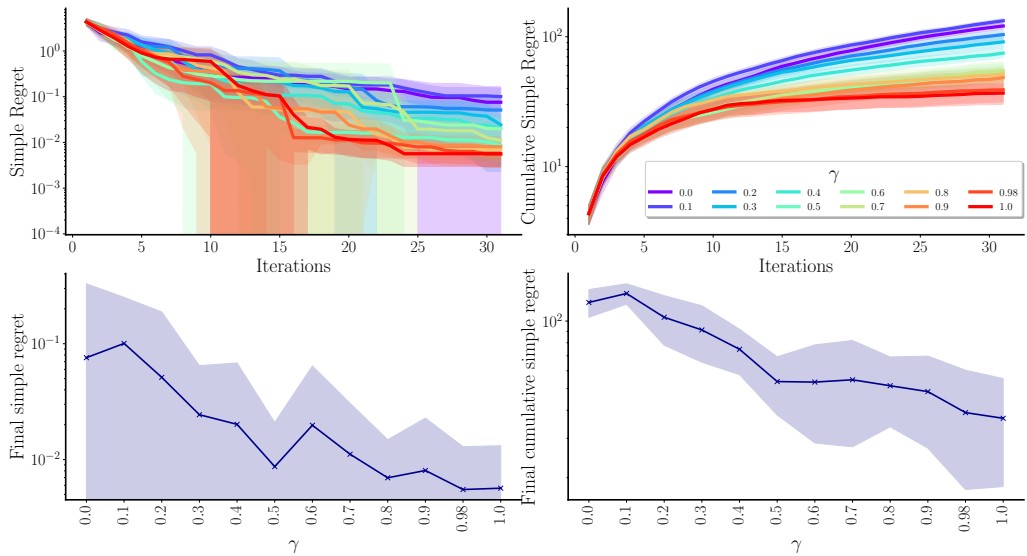

Figure S8: Ablation Study on the discount factor $\gamma$ (Equation 3) on the 1D Forrester function, with a query set size $S = 256$. Mean with 95% confidence intervals computed across 30 runs with random starting pairs. **As expected, an higher value of $\gamma$ leads to a lower cumulative simple regret. The picture is more nuanced for simple regret, even though as we progress towards a large number of iterations, the conclusion appears to be similar.**

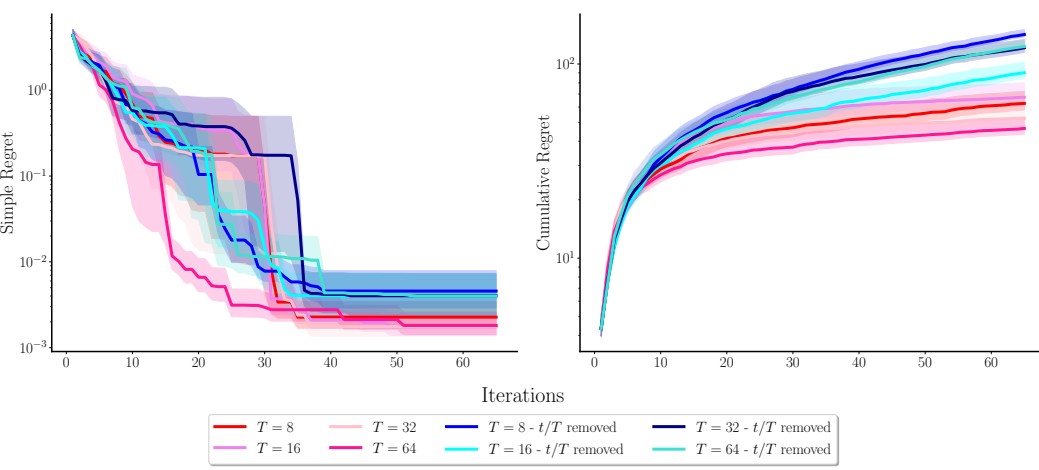

Figure S9: Simple regret and cumulative simple regret on the 1-dimensional Forrester function depending on the query budget $T$ and whether or not $t/T$ is passed as input to the acquisition head $f_a$. Mean with 95% confidence intervals computed across 30 runs with random starting pairs. **The variance is mostly driven by whether or not $t/T$ is passed to the acquisition head or not (see Figure 2, right). When passing $t/T$, a large budget $T$ is associated with the fastest convergence and the lowest simple cumulative regret.**

## B.4 EXAMPLE OF INFERENCE FUNCTION SHAPE AND OPTIMUM ON TEST FUNCTIONS

To provide qualitative insight into PABBO's learning capabilities, Figure S11 visualizes the function landscapes learned by our model after 30 optimization steps on two test functions. For the 2D Branin function (left), PABBO successfully identifies one of the three global optima. For the 1D Forrester function (right), PABBO accurately captures both the overall shape and the location of the global optimum.

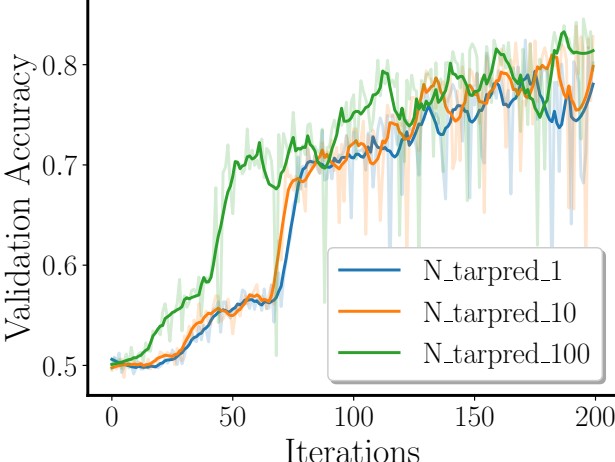

Figure S10: Validation accuracy on 128 1-dimensional GP samples. **Increasing the size of $D^{\text{(tarpred)}}$ provides more loss terms during training, thereby enriching the training signal, and resulting in faster convergence of the network.**

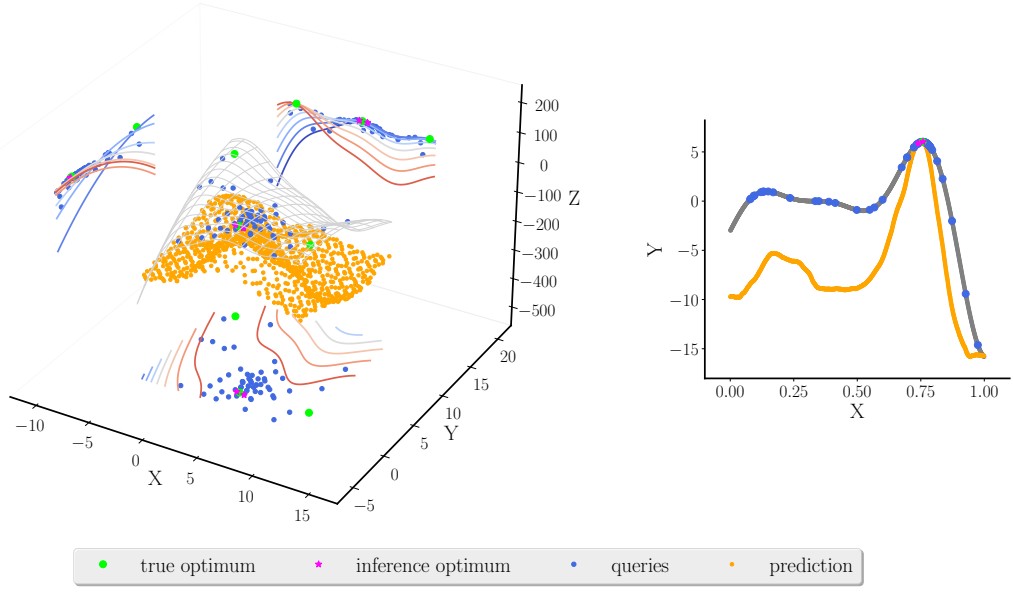

Figure S11: Optimum inference and function shape on test functions after 30 optimization steps for the $2D$ Branin function (left) and $1D$ Forrester function (right) trialed in Section 5.1.

## C    GENERATIVE PROCESS OF SYNTHETIC DATASET

To generate the $D-$dimensional synthetic data in Section 5.1, we first sample from a GP, where

- The kernel are equally sampled from the RBF, Matérn-$1/2$, Matérn-$3/2$ and Matérn-$5/2$ kernels, with the kernel standard deviation $\sigma \sim \mathcal{U}([0.1, 2])$ and lengthscale for each dimension $l^d \sim \mathcal{N}(1/3, 0.75)$ truncated to $[0.05, 2]$.

- Then we sample function mean as the maximal value of $m$ observations from the Normal distribution $\mathcal{N}(0, \sigma^2)$. To account for very high optimum, $\exp(1)$ is added to the mean with a probability of $0.1$.

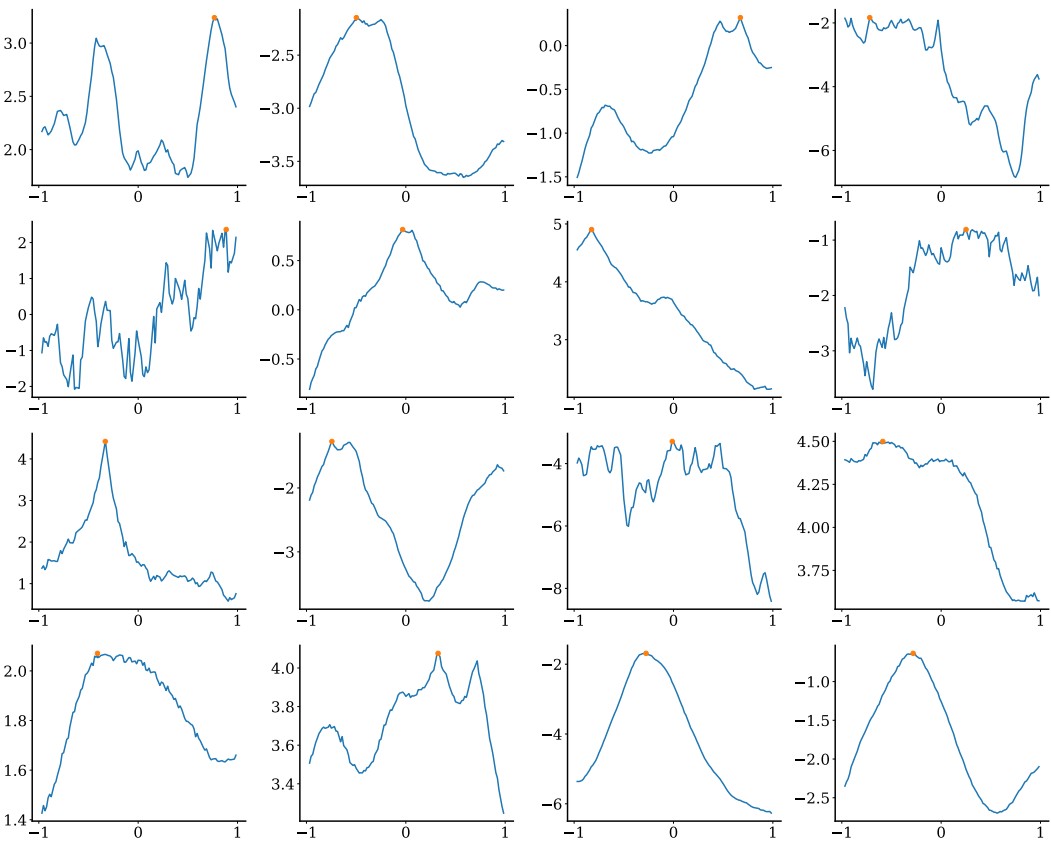

Figure S12: Example of 1-dimensional synthetic data generated using the protocol defined in Appendix C. The orange dot in each plot indicates the optimum of the sampled function.

- With the defined prior, we randomly sample an optimum $(\boldsymbol{x}^*, 0)$ inside $[-1, 1]^D$ and a total of $N - 1$ context points, where $N = 100 \times D$. The context set is sampled from GP posterior conditioned on the optimum by first sampling $M - 1$ points conditioned on the optimum, and the rest of $N - M$ points conditioned on these points and the optimum. $M = 50$ for 1-dimensional data and 100 when $D = 2, 3, 4$.

- To ensure the existence of global optimum, we additionally add a quadratic bowl at $\boldsymbol{x}^*$.

- Finally, the function value is $y = -(|y| + \frac{1}{8}\|\boldsymbol{x}^* - \boldsymbol{x}\|^2 + \Delta y)$. The offset $\Delta y \sim \mathcal{U}([-5, 5])$, makes the maximal value $y^* = (0 + \Delta y) \in [-5, 5]$.

Figure S12 shows 16 randomly generated 1-dimensional synthetic functions.

# D TRAINING AND EXPERIMENTAL DETAILS

## D.1 PABBO HYPERPARAMETERS

| Model Architecture | |
|---|---|
| Embedding dimension of Transformer | 64 |
| Point-wise feed-forward dimension of Transformer | 128 |
| Number of self-attention layers in Transformer | 6 |
| Number of self-attention heads in Transformer | 4 |
| Number of hidden layers in data embedders | 3 |
| Number of hidden layers in decoder ($f_p, f_a$) | 1 |
| Hidden layer dimension of decoder ($f_p, f_a$) | 128 |
| **Training** | |
| Number of iterations for warm-up | 3000 |
| Total number of training iterations | 8000 |
| Horizon of episodes ($T$) | 64 |
| Learning rate for warm-up | $1 \cdot 10^{-3}$ |
| Learning rate after warm-up | $3 \cdot 10^{-5}$ |
| Learning rate decay | Cosine decay to 0 over 8000 iterations |
| Number of meta-tasks in a batch during warm-up ($B$) | 128 |
| Number of meta-tasks in a batch after warm-up ($B$) | 16 |
| Number of trajectories from one meta-task for policy learning | 20 |
| Weight on auxiliary loss ($\lambda$) | 1.0 |
| discount factor ($\gamma$) | 0.98 |
| Size of query set ($S$) | $\min(300, 100 \cdot D)$ |
| Size of candidate query pair set ($M$) | $\min(300, 100 \cdot D)$ |
| Size of prediction set ($N$) | $\min(300, 100 \cdot D)$ |
| Maximal size of prediction context set ($N^{(\text{ctxpred})}$) | $50(D=1), 100(1 < D \le 4), 200(D > 4)$ |
| **Environment** | |
| Number of initial pairs during training | 0 |
| Number of initial pairs during evaluation | 1 |
| Observation noise of duel feedback ($\sigma_{\text{noise}}$) | 0.0 |

Table S1: Hyperparameter settings used for PABBO. Most of them remain consistent across all the tasks. Additionally, the size of meta-datasets was scaled according to the search space dimension.

## D.2 HPO DATASET DESCRIPTION AND EXPERIMENTAL DETAILS

As mentioned in Section 5.2 and Appendix B.2, we experimented on four high-dimensional search spaces from the HPO-B benchmarks (Pineda Arango et al., 2021), each corresponding to the hyper-paramers of a certain model: No. 5636 for `rpart(29)`, No. 5859 for `rpart(31)`, No. 7609 for `ranger(16)`, and No. 5971 for `xgboost(6)`. Both No. 5636 and No. 5859 are 6-dimensional spaces, No. 7609 is a 9-dimensional space, and No. 5971 is a 16-dimensional problem. Meta-datasets within each space are divided into three splits: meta-train, meta-validation, and meta-test.

An individual model is trained on the meta-train split of each search space. Each meta-dataset is equally divided beforehand into two parts from which we sample either prediction set $\mathcal{D}^{(p)}$ or query set $D^{(q)}$, so as to prevent any information leak from rewards. The only exception happens when a meta-dataset has too few data points: we fit a GP to the original data points and sample from the posterior to generate either $\mathcal{D}^{(p)}$ or $\mathcal{D}^{(q)}$. As described in Section 3.2, we sample $\mathcal{D}^{(p)}$ containing $N$ queries and preferences and $\mathcal{D}^{(q)}$ with $S$ query points from each meta-dataset at the beginning of each training iteration.

## D.3 DETAILS OF BASELINES

The PBO baselines are implemented based on `PairwiseGP` model class from `BoTorch`. For qEUBO, qEI and qNEI, `BoTorch` provides ready-to-use acquisition class `qExpectedUtilityOfBestOption`, `qExpectedImprovement` and `qNoisyExpectedImprovement`. For qTS, we sample two draws from the GP posterior at: (1) locations from quasi-random Sobol sequences for continuous search spaces, or (2) locations of all candidates for discrete search spaces. The maximum of each draw are chosen as the

next query. For MPES, we follow the implementation based on (Nguyen et al., 2020), which is available at `https://github.com/RaulAstudillo06/qEUBO/tree/main/src/acquisition_functions`. Finally, we select a pair of random points for continuous spaces and a random pair of possible candidates for discrete spaces as a random strategy. Baseline inputs are all normalized to $(0, 1)$.

## D.4 HARDWARE

We train our model using up to 5 Tesla V100-SXM2-32GB GPUs, with training time of roughly 10, 25, and 29 hours for 1-, 2-, and 4-dimensional synthetic data respectively. There is significant potential to shorten this pre-training process by creating the training samples beforehand, rather than generating them online. For the HPO-B tasks on search spaces No. 5636 and No. 5859 (6 dimensions) and the search space No. 7609 (9 dimensions), training takes approximately 21.5 hours per task. We evaluate all the models on 2x64 core AMD EPYC 7713 @2.0 GHz.

