# OpenReview forum: "PABBO: Preferential Amortized Black-Box Optimization"
_ICLR.cc/2025/Conference — ICLR 2025 Spotlight_

### Official Review · Reviewer_do2x · 2024-10-30

**Soundness:** 3
**Presentation:** 3
**Contribution:** 2
**Rating:** 5
**Confidence:** 4

**Summary:**

This paper introduces the Preferential Amortized Black-Box Optimization (PABBO) approach, which learns acquisition function values for candidate pairs based on preferential feedback between pairs of designs. The method utilizes a transformer-based architecture and a reinforcement learning-based pretraining scheme.

**Strengths:**

The PABBO architecture achieves end-to-end training for preference feedback and employs an application of amortization within a preferential Bayesian optimization (PBO) setting. The algorithm’s contributions are demonstrated through experiments against Gaussian process-based benchmarks.

**Weaknesses:**

This paper seems to fall below the standard expected at ICLR for the following reasons:

1. A critical issue is that the paper lacks a theoretical foundation or even a mathematical intuition for its approach, focusing primarily on reporting numerical results without a deeper analytical context.

2. In terms of experiments, the explanation of the architecture design and the choice of hyperparameters requires more clarity and justification. I will outline these concerns in further detail in the questions below.

3. Finally, the paper would benefit from additional polishing. For instance, in line 189, y_{i,1} and y_{i,2} appear without prior explanation. In line 219, the term “P” in “the rest of P queries” may be confusing for readers. In line 718, For in "M=50 For" should be for.

**Questions:**

About the architecture:
1. Why did you choose the Gaussian distribution in equation (4)? What's the role of the Gaussian distribution in the algorithm's success for the out-of-distribution case in synthetic experiments of section 5.1 and human preferences in section 5.3? Are there any mathematical intuitions?
2. What's the influence of D^{ctxpred} in your training procedure? How do you determine the percentage of D^{ctxpred} and D^{tarpred}?

About the hyperparameters:
1. Why do you fix lambda=1 in all experiments? What's the influence of lambda?
2. In line 500, why do you choose gamma=0.5, 0.9, 0.98, 1.00? These choices seem unusual and may have been selected deliberately, potentially compromising the generality of the results.
3. What's the influence of query budget T?
4. What would high dimensions influence the algorithm? For example, what would happen if you chose the D to be very large in line 719?

I would be willing to increase the score if the above questions could be well clarified.

---

> ### Author Response · Authors · 2024-11-19
> **Answer 1/1**
>
> Comments that can be answered through additional experiments will be posted as soon as possible when the latter are completed. With respect to the reviewer's comments, these include:
>
> - Consider higher dimensional examples.
> - Performing an ablation study on $\lambda$ in the computation of the loss $\mathcal{L} = \mathcal{L}^{(q)} + \lambda\mathcal{L}^{(p)}$.
> - Performing an ablation study on the discount factor $\gamma$ (Equation 3)
> - Performing an ablation study on the query budget $T$ used during pre-training.
> - Performing an ablation study on the percentage of $D^{\text{ctxpred}}$ and $D^{\text{tarpred}}$
> Additional comments can be found in the global comment.
>
> **A critical issue is that the paper lacks a theoretical foundation or even a mathematical intuition for its approach, focusing primarily on reporting numerical results without a deeper analytical context.**
>
> We would like to begin by emphasizing that unlike the BO field, PBO itself has long been functioning without formal theoretical guarantees. These are made cumbersome by the preferential nature of the feedback, which entails that the black-box function can only be identified up to a monotonic transformation. The first results are only a few months old: [1] established asymptotic consistency for the Dueling Thompson Sampling acquisition function for a finite query set, and [2] provided the first regret bound. Such theoretical results make strong assumptions about the black-box function, which in turn requires care when selecting a GP surrogate kernel and acquisition strategy. PABBO avoids these considerations by directly learning the acquisition function in a data-driven manner, at the loss of theoretical results.
>
> Nevertheless, we can provide some mathematical justification. At the center of our method is Transformer Neural Process (TNP) [3], a well-suited model class for PBO due to several key properties:
>
> - The optimization history in PBO is inherently a *set* of preference pairs, meaning that the order of observations should not affect the acquisition decision. TNPs naturally handle this through self-attention mechanisms, maintaining permutation equivariance in the intermediate layers and invariance in the final output.
> - The attention mechanism allows adaptive computation where each preference pair's representation is influenced by the entire optimization history. This is crucial for PBO as the relative importance of each pair varies depending on the current stage of optimization.
> - TNPs inherit the universal approximation capabilities of transformers [4], and can learn arbitrary continuous mappings from sets of preferences to acquisition values, allowing direct learning of a joint surrogate model and acquisition function in an end-to-end manner.
>
> **We will briefly mention some of these mathematical insights in the main text, in section 3.3.**
>
> **Why did you choose the Gaussian distribution in equation (4)?**
>
> We followed the common practice in the neural processes literature of using independent (diagonal) Gaussian likelihoods [5]. If modeling correlations between points is crucial for the downstream task, we can replace the output with a joint multivariate normal distribution (similar to GNP [6]) or predict the output autoregressively (like AR-CNP [7]). For modelling multimodal predictive distributions, we could replace the Gaussian head with a mixture-of-Gaussians head. These modifications can be easily implemented.
> Additionally, in PBO, a likelihood often used is the probit likelihood, which assumes Gaussian noise [1, Section 3.1]. As our goal is to amortize PBO, a Gaussian noise model is a relevant assumption. **A short sentence summarizing this answer will be added as a justification for Equation 4 in the main text.**
>
> [1] Astudillo et al. Preferential Multi-Objective Bayesian Optimization.
>
> [2] Xu et al. Principled Preferential Bayesian Optimization
>
> [3] Nguyen \& Grover. Transformer neural processes: Uncertainty-aware meta learning via sequence modeling.
>
> [4], Yun et al. Are transformers universal approximators of sequence-to-sequence functions?
>
> [5] Garnelo et al. Conditional neural processes.
>
> [6] Markou et al. Practical conditional neural processes via tractable dependent predictions.
>
> [7] Bruinsma et al. Autoregressive conditional neural processes.
>
> [8] Brochu et al., A Tutorial on Bayesian Optimization of
> Expensive Cost Functions, with Application to
> Active User Modeling and Hierarchical
> Reinforcement Learning

---

> > ### Author Response · Authors · 2024-11-23
> > **Responses related to additional experiments**
> >
> > We now address all remaining questions related to the experiments.
> >
> > **Consider higher dimensional examples.**
> >
> > We have carried out an additional experiment on a 16-dimensional HPO-B task (Figure 10). Please refer to our response to reviewer Tep5 for further details.
> >
> > **Ablation study on $\lambda$, in the computation of the loss $\mathcal{L} = \mathcal{L}^{(q)} + \lambda\mathcal{L}^{(p)}$.**
> >
> > We conducted an ablation study to evaluate the impact of the loss weight $\lambda$ and the presence of a warm-up phase (L296) on PABBO's performance. Our results, presented in Figure 8, show that the warm-up phase is indeed critical for improving the model's performance. We also observe that varying the loss weight $\lambda$ has only a mild effect on the overall performance, both in the presence and absence of the warm-up phase. This indicates that while the prediction task plays a key role in training, the exact weighting of $\mathcal{L}^{(p)}$ is not highly sensitive. As mentioned in the paper, we believe the role of this prediction loss in the warm-up phase is to *stabilize* training: by first learning the shape of the latent function, we reduce erratic behaviors during policy training, by reducing the strong coupling between the shape of the function and the acquisition values.
> >
> > **Ablation study on the discount factor $\gamma$ (Equation 3).**
> >
> > To address your concern, we have conducted a more comprehensive ablation study by considering a denser grid of $\gamma$ values, spaced between 0 to 1, with results shown in Figure 9. The conclusion aligns with the main text: smaller $\gamma$ values prioritize immediate rewards, leading to slightly faster convergence in the early stage but worse performance in the long term. Additionally, we observe that $\gamma < 0.5$ deteriorates performance, our guess is that sparse reward signals hinder the model convergence.
> >
> > Based on these findings, we recommend using a $\gamma$ close to 1 when the primary goal is minimizing final regret. For scenarios where early-stage performance is crucial, a $\gamma$ between 0.5 and 1.0 can be considered.
> >
> > **Ablation study on the query budget $T$ used during pre-training.**
> >
> > We carried out an ablation study to evaluate both the impact of varying $T$ and the effect of injecting the temporal information $t/T$ into the neural network, the results are shown in Figure 13. Our results indicate that including $t/T$ as input to the acquisition head improves the performance, which means the temporal information $t/T$ helps the model better capture the progression of the optimization process, leading to more informed acquisition decisions. Additionally, we observe that when $t/T$ is included, a larger query budget $T$ is associated with faster convergence and the lowest cumulative simple regret.
> >
> > **Ablation study on the percentage of $D^{\text{ctxpred}}$ and $D^{\text{tarpred}}$.**
> >
> > In our experiments, rather than fixing a specific ratio between $D^{\text{ctxpred}}$ and $D^{\text{tarpred}}$, we use a dynamic scheme. We set a total number of samples for the prediction set and randomly sample $D^{\text{ctxpred}}$ and assign the remaining pairs as $D^{\text{tarpred}}$. The dynamic adjustment allows the neural network to accurately estimate the predictive posterior distribution regardless of the size of the context set.
> >
> > As for the size of $D^{\text{tarpred}}$, increasing its size provides more loss terms during training. Specifically, in the neural process framework, having more targets allows the model to estimate the log-likelihood over a larger number of targets, thereby enriching the training signal. This will result in faster convergence of the network. We conducted an ablation study to validate this effect, with results shown in Figure 14.
> >
> > When computational resources allow and sufficient data is available, using a larger $D^{\text{tarpred}}$ is beneficial, as it speeds up convergence and improves the overall training efficiency of the model.
> >
> > **We have now answered all of your questions and concerns.** Additionally, in response to feedback from other reviewers, we have included new baselines, additional experiments, and other ablation studies. A detailed summary of these updates can be found in Global Comment 2 and in our responses to other reviewers. Given these new results, we sincerely hope the reviewer might reconsider their assessment.

---

> > > ### Comment · Reviewer_do2x · 2024-11-26
> > > **Increase the score**
> > >
> > > Thanks to the author for their response and explanation. Based on the changes, I decided to increase my score to 5.

---

> > > > ### Author Response · Authors · 2024-11-27
> > > > **Thank you!**
> > > >
> > > > We sincerely appreciate your recognition of our revisions and thank you for increasing the score. We
> > > > are happy to address any further concerns you may have.

---

### Official Review · Reviewer_x3e8 · 2024-11-02

**Soundness:** 3
**Presentation:** 3
**Contribution:** 3
**Rating:** 8
**Confidence:** 3

**Summary:**

This paper introduces a new approach to preferential Bayesian Optimization, using large-scale pretraining to enhance inference speed and optimization performance. PABBO employs two distinct transformer heads: a Prediction Head that captures the underlying function determining preferences, and an Acquisition Head that selects the next query pair. Experimental results show that PABBO outperforms existing methods in accuracy and computational efficiency.

**Strengths:**

1. The idea of utilizing meta-learning for Amortized Optimization in preferential Bayesian Optimization is new and reduces the cost of inference.

2. Experiment results show that PABBO outperforms other methods in the datasets selected in this paper.

**Weaknesses:**

1. The pretraining in this method requires a large amount of data, which could be costly for certain tasks.

2. PABBO is currently limited to tasks with a fixed input dimensionality.

3. PABBO only supports pairwise preference queries, limiting its ability to handle queries with multiple options, which could reduce the efficiency of information gathering. While existing works like [1] could handle queries with multiple options.

4. Including more datasets in the experiments would provide a more comprehensive evaluation. For example, incorporating all acquisition functions and tasks from [1] and other related works could strengthen the comparison.

Reference:
1. Astudillo, R., Lin, Z.J., Bakshy, E. and Frazier, P., 2023, April. qEUBO: A decision-theoretic acquisition function for preferential Bayesian optimization. In International Conference on Artificial Intelligence and Statistics (pp. 1093-1114). PMLR.

**Questions:**

NA

---

> ### Author Response · Authors · 2024-11-19
> **Answer 1/1**
>
> Comments that can be answered through additional experiments will be posted as soon as possible when the latter are completed. With respect to the reviewer's comments, these include:
>
> - Considering additional test functions, such as those used in Astudillo *et al.*, qEUBO: A decision-theoretic acquisition function for preferential Bayesian optimization.
>
> Additional comments can be found in the global comment.
>
> **The pretraining in this method requires a large amount of data, which could be costly for certain tasks.**
>
> We are not sure if by ''costly'', the reviewer is referring to the time-consuming pre-training on large amounts of data, or to the cost linked to gathering such large amounts of data. We address both concerns below:
>
> - For the former, by definition of Amortized Inference, one trades a long but offline pre-training time for fast inference at test time. If the model is to be used on many related tasks afterward, this is a highly beneficial strategy. Typically, for optimizing human preferences, the speedup obtained at inference time allows for seamless interaction with users.
> - For the latter, we would assume that a pre-training dataset, containing the numerous optimization runs done before, would be available from the start. Most importantly, the dataset can feature synthetic examples, e.g. generated with draws of many different Gaussian Processes, as we outlined in Appendix A.1. Our empirical results demonstrated the soundness of this approach. Leveraging synthetic data from amortized inference is now a widely used practice for diverse tasks such as time-series prediction [1, Section 4], mixed-effect models inference [2], and, more to the point, Bayesian Optimization [3, Section 5.2 and Appendix E].
>
> **PABBO is currently limited to tasks with a fixed input dimensionality.**
>
> This is a current bottleneck of the approach, as PABBO needs to be retrained from scratch when switching between tasks of different input dimensionality. The whole amortized optimization field is facing this challenge. There has now been recent work [4, 5, 6] to tackle this issue. Shortly, the idea is to utilize a bi-dimensional attention mechanism that can accept data with different dimensions.
> At the moment, integrating this new architecture within PABBO represents one of the most relevant avenues for future work. Leveraging such approaches in black-box optimization holds great potential, as one could use knowledge acquired on previous lower-dimensional tasks to inform high-dimensional tasks.
>
> **PABBO only supports pairwise preference queries, limiting its ability to handle queries with multiple options, which could reduce the efficiency of information gathering. While existing works could handle queries with multiple options.**
>
> We agree with the reviewer. Our architecture could be extended to contain as many data embedders as there are points to compare. However, preferences over $q>2$ queries ultimately reduce to $\frac{q(q-1)}{2}$ pairwise preferences. Given the speedup provided by PABBO, this alternative remains amenable when interacting with humans.
>
> [1] Ansari et al. Chronos: Learning the Language of Time Series.
>
> [2] Arruda et al. An amortized approach to non-linear mixed-effects modeling
> based on neural posterior estimation.
>
> [3] Huang et al. Practical Equivariances via
> Relational Conditional Neural Processes.
>
> [4] Anonymous. Dimension Agnostic Neural Processes. https://openreview.net/forum?id=uGJxl2odR0
>
> [5] Dutordoir et al. Neural diffusion processes.
>
> [6] Liu et al. Task-agnostic amortized inference of Gaussian process hyperparameters.

---

> > ### Comment · Reviewer_x3e8 · 2024-11-20
> >
> > Thank you for your response. Since this article focuses on empirical results, comparison with the baseline is particularly important. Therefore, I will maintain my score until further experimental results are completed.

---

> > > ### Author Response · Authors · 2024-11-23
> > > **Responses related to additional experiments**
> > >
> > > We now have implemented an additional baseline, MPES [1], which is one of the baselines being compared in qEUBO [2]. The results are presented in Figure 11. In a similar spirit, we added the Ackley 6D function (Figure 12), which is also part of the evaluations carried out in [2]. On all these synthetic examples, PABBO outperforms MPES. Of note, the latter requires inference times that are an order of magnitude above other GP-based strategies.
> > >
> > > Regarding the Ackley function, let us mention that, to address the concerns regarding query set size scalability (Tep5), we included a ''batch'' version of PABBO, where, instead of passing a query set of size $S$, we pass a batch of $B$ query set of size $S$. This increases the effective query set size while preserving fast inference times, and results in faster convergence.
> > >
> > > In addition to the above, we have also included a new high-dimensional experiment and conducted a series of ablation studies to address various reviewer concerns. A detailed summary of these new experiments and findings can be found in Global Comment 2 and in our responses to other reviewers. We have made significant efforts to strengthen the empirical evaluation of PABBO, ensuring a more comprehensive comparison and deeper insights into its performance. Given these additional results, we sincerely hope the reviewer might reconsider their assessment of the paper.
> > >
> > > [1] Nguyen et al. Top-k ranking Bayesian optimization.
> > >
> > > [2] Astudillo et al. qEUBO: A decision-theoretic acquisition function for preferential Bayesian optimization.

---

> > > > ### Comment · Reviewer_x3e8 · 2024-11-25
> > > >
> > > > Thank you for your response. I noticed that the qEUBO paper includes experiments on tasks like Alpine1, Hartmann, Animation, and Carcab. Have you tried your method on these tasks as well? Also, the qEUBO paper mentions the qNEI method—did you compare your approach with qNEI? If possible, it would be great to see the experiment results for these tasks and methods to better understand how your method performs in comparison.

---

> > > > > ### Comment · Reviewer_x3e8 · 2024-11-26
> > > > >
> > > > > I have updated my score to 6. I may consider increasing it further if additional experimental results are provided.

---

> > > > > > ### Author Response · Authors · 2024-11-27
> > > > > > **Additional set of experiments: qNEI and Hartmann6D**
> > > > > >
> > > > > > We sincerely thank you for reconsidering our work and increasing the score to 6.
> > > > > >
> > > > > > To address the reviewer’s concerns: evaluating PABBO on the suggested functions would require
> > > > > > training new models from scratch, as PABBO is not dimension-agnostic (a future direction discussed
> > > > > > in Section 6). Moreover, running the GP-based baselines on these functions requires significant
> > > > > > computation time, making it difficult to obtain results in time for this rebuttal. However, we had
> > > > > > already trained a 6D PABBO model for Ackley6D, allowing us to experiment Hartmann6D. We
> > > > > > also added the suggested qNEI baseline to our benchmarks on synthetic functions. **The PDF now
> > > > > > includes qNEI in Figures 11 and 12. Figure 13 reports results on Hartmann6D**.
> > > > > >
> > > > > > In our experiments, qNEI slightly outperforms PABBO only on GP1D, and performs on par for
> > > > > > GP2D. On other functions (Forrester, Beale, Branin, Ackley, Hartmann), qNEI does not outperform
> > > > > > PABBO. Additionally, qNEI incurs longer inference times than all GP-based methods, except MPES.
> > > > > >
> > > > > > Regarding Hartmann, PABBO does not match the performance of qEUBO and qEI. Upon in-
> > > > > > vestigation, it appears that the Hartmann function is quite different from our pre-training dataset,
> > > > > > and involves many bumps, similar to a periodic function. This suggests that a more diverse pre-
> > > > > > training dataset, including periodic kernels, could help mitigate this issue (See our ablation on the
> > > > > > pre-training dataset composition in Figure 6 and our response to Tep5). However, due to time
> > > > > > constraints, we cannot retrain the model at this stage but will consider this in future work. We
> > > > > > have already acknowledged the limitations of our pre-training in the paper, and this example further
> > > > > > highlights the need for additional kernels, such as periodic or nonstationary ones.
> > > > > >
> > > > > > We believe our experimental validation is now comprehensive and representative, with a bench-
> > > > > > mark covering 13 diverse problems: 7 synthetic functions, 4 HPO search spaces, and 2 human
> > > > > > preferences datasets, ranging from 1D to 16D. PABBO achieves the lowest simple regret overall and
> > > > > > does so with reduced inference times. Qualitatively, across all 13 problems, PABBO has an average
> > > > > > rank of 1.5 for simple regret. **Notably, on 6 problems, PABBO ranks first with a statistically
> > > > > > significant margin**. The second-best baseline, qNEI, ranks 3rd on average, but only on the 8
> > > > > > synthetic problems considered through Figures 11-12-13. It never wins by a statistically significant
> > > > > > margin. Nevertheless, given that qNEI outperforms other GP-based strategies, adding it to the
> > > > > > benchmark strengthened our evaluation. When considering all 13 test functions, qEI is the second-
> > > > > > best performer, with an average rank slightly above 3. Given the consistent performance across a
> > > > > > diverse set of problems, adding more test functions is unlikely to change the conclusions.
> > > > > > We hope that this new set of experiments, which added a test function and a baseline, addresses
> > > > > > the concerns raised by the reviewer.

---

> > > > > > > ### Comment · Reviewer_x3e8 · 2024-11-28
> > > > > > >
> > > > > > > Thank you for providing the additional experiments and detailed explanations. I now consider the paper both novel and meaningful. While there are some limitations (such as PABBO not being dimension-agnostic and requiring new models to be trained for functions with different dimensions), these are understandable and could be addressed in future work. I’ve decided to raise my score to 8.

---

> > > > > > > > ### Author Response · Authors · 2024-11-28
> > > > > > > >
> > > > > > > > Thank you for recognizing our additional efforts and increasing your score, we really appreciate it.

---

### Official Review · Reviewer_dRx5 · 2024-11-04

**Soundness:** 3
**Presentation:** 4
**Contribution:** 3
**Rating:** 8
**Confidence:** 4

**Summary:**

This paper presents PABBO a preference based Bayesian optimization approach using transformer architectures and deep sets approaches to amortize the inference time. The paper extend ideas of amortization from past works to preference based optimization and presents an end-to-end trainable approach.

The main idea behind the approach is to use a transformer and attention mechanisms on historical observed data and use them to make query predictions. Another prediction head from the same transformer produces the policy function which is used to suggest the future queries. Experimental results indicate that the proposed approach is faster compared to previous approach and achieves a smaller regret.

**Strengths:**

Strengths:
- The paper presents a new approach to preference based BO using transformers which as several advantages - it can incorporate preferences, has amortized inference cost and can be trained end-to-end.
- The proposed approach is intuitive and a reasonable way to model preferences in BO, with the potential to scale to very large problems. The method uses neural networks instead of Gaussian processes which helps it scale to a large number of data points.
- Extensive experimentation shows that PABBO has a smaller inference cost while achieving a smaller regret compared to prior approaches.

**Weaknesses:**

Weaknesses:
- While the proposed approach is faster than BO, the scale at which the experiments are performed (100 observations max), the gains are not fully realized since BO methods can perform quite comfortable at such small scales.
- The full potential of the algorithm is only realized in large experiment with thousands of observations. Such large scale experiments have not been presented in the paper.
- Preference based optimization is most useful when modeling users with varying preferences. The paper can benefit from the interesting extension of learning to model individual user preferences.

**Questions:**

General questions
- How does preference based optimization work when users have varying preferences? Is it possible to incorporate personal user-preferences in this framework.

Specific questions
- Line 237, Is the same MLP applied to x_i,1, x_i,2 and l_i individuall? Why should l_i be encoded into the same embedding space as x_i,1 and x_i,2? Or does the statement mean to encode the concatenation of the triple using an MLP?
- Line 267, It seems that to train $\pi_\theta$ the reward used is given in line 305 defined as the maximum of the observations so far. This part is unclear to me.
  - How is this reward helping in learning a good policy?
  - It seems that the loss function is static given a history H_t, so it is not a RL setup but simply a supervised learning setup. Is any RL specific learning procedure being used here? Are the reward values r_t fixed observations or are they learnable and also backpropagated through to the prediction values y and consequently through the prediction head?
  - This step looks very similar to offline BO where a offline data set is used to learn an acquisition function or a function approximation. What is the relationship of this method to offline BO?
- Line 303, How is the query set constructed when computing $\pi_\theta$? Is it a random subset of all query pairs? Or is any specialized strategy used to locate the most promising queries?
- How is PABBO able to avoid sub-optimal solutions caused due to not exploring the whole optimization space?

---

> ### Author Response · Authors · 2024-11-19
> **Answer 1/2**
>
> Comments that can be answered through additional experiments will be posted as soon as possible when the latter are completed. With respect to the reviewer's comments, these include:
>
> - Carry out experiments involving a higher number of iterations.
>
> Additional comments can be found in the global comment.
>
> **How does preference based optimization work when users have varying preferences? Is it possible to incorporate personal user-preferences in this framework.**
>
> We ask for clarifications: is the reviewer talking about handling multiple users, each with distinct preferences, or about a single user whose preferences evolve over time, as in nonstationary Bayesian Optimization [1]? The former case is naturally handled by our framework; different users are simply considered as different tasks that are assumed to share some similarities with those seen during pre-training. The latter is more complex and is currently not accounted for by our method, even though PABBO can be readily applied to this setting. Investigating suitable training procedures for that scenario as well as dedicated pre-training datasets constitutes a relevant avenue for work if one wants to further advance the field of human preference learning from an amortized perspective.
>
> **Line 237, Is the same MLP applied to $x_{i,1}, x_{i,2}$ and $l_i$ individually? Why should $l_i$ be encoded into the same embedding space as $x_{i,1}$ and $x_{i,2}$? Or does the statement mean to encode the concatenation of the triple using an MLP?**
>
> To clarify, PABBO employs *separate* MLPs for $x_{i, 1}, x_{i, 2}$ and $l_i$, mapping each of them to *separate* embeddings of
>  Transformer input dimension as $\textbf{E}^{x_{i, 1}}$, $\textbf{E}^{x_{i, 2}}$,  and $\textbf{E}^{(l_i)}$, with $\oplus$ denoting the component-wise addition.
> When working on the context set $\mathcal{D}^{(c)}$ and prediction context set $\mathcal{D}^{\text{(ctxpred)}}$, which consists of duels $(x_{i, 1}, x_{i, 2}, l_i)$,  the token embedding is obtained by adding the individual embeddings up, i.e. $\textbf{E} = \textbf{E}^{x_{i, 1}}\oplus \textbf{E}^{x_{i, 2}}\oplus\textbf{E}^{(l_i)}$. When processing a single design from the query set $\mathcal{D}^{(q)}$ or prediction target set $\mathcal{D}^{\text{(tarpred)}}$, we directly use the individual embedding for that design as the token embedding, i.e., $\textbf{E} = \textbf{E}^{x_{i, 1}}$. Because $(x_{i,1}, x_{i, 2}, l_i)$ forms a single atomic element in our context set, Transformer architecture requires all elements in its input sequence to live in the same embedding space: to pass these preference observations to the transformer, we need to make sure their dimensions are aligned.
>
> Given that PABBO encodes elements separately, for the same type of elements, like $x$, a shared embedder can be used across all three sets. This enables parameter sharing for the embedders, as the same embedder processes all $x$-type input, regardless of which set they belong to. This reduces model complexity because it avoids the need to train separate embedding functions for each set. If inputs were concatenated to obtain a joint embedding, this ability to reuse a shared embedded for the same type of elements would be lost, resulting in redundant parameters, since the
> embedder would need to relearn shared patterns for each set independently.
> Finally, separate embeddings confer distinct semantic meanings to features $x$ and preference labels $l$. Thus, separate MLPs can specialize in their respective tasks: Feature embedders can focus on capturing spatial relationships, and preference embedders can focus on binary relationship encoding. This leads to an increased representation power. For all these reasons, we restrained from employing concatenation to obtain a joint embedding.
>
> This being said, a legitimate question to ask now is ''why not use the same embedding for $x_{i,1}$ and $x_{i,2}$''. Basically, in this case, upon summing $\textbf{E}^{x_{i, 1}}$ and $\textbf{E}^{x_{i, 2}}$ the order information would be lost: the network would not know whether $x_{i,1} > x_{i,2}$, or the opposite. That is why we used separate embedders for $\textbf{E}^{x_{i, 1}}$ and $\textbf{E}^{x_{i, 2}}$.
>
> These details were missing from the main text, and we thank the reviewer for helping us identify them. As a result, **we will modify the main text, briefly elaborating on the benefits of addition and separate embedders.**
>
> [1] Deng et al., Weighted Gaussian Process Bandits for Non-stationary Environments.

---

> > ### Author Response · Authors · 2024-11-19
> > **Answer 2/2**
> >
> > **Line 267, It seems that to train
> >  the reward used is given in line 305 defined as the maximum of the observations so far. This part is unclear to me.**
> >
> > - **How is this reward helping in learning a good policy?**
> >
> >  The reward represents the maximum utility observed so far based on the latent function, whose values are always available during pre-training. It quantifies the **quality of query pairs suggested by the query head** in terms of their contribution to optimizing the underlying function. This reward guides the policy gradient method (Equation 3) to improve the query head, ensuring it proposes pairs that are more likely to identify regions of high utility. By using cumulative discounted rewards across the optimization trajectory, the policy is trained to balance immediate gains with exploration.
> >
> > - **It seems that the loss function is static given a history $H_t$, so it is not a RL setup but simply a supervised learning setup. Is any RL specific learning procedure being used here? Are the reward values $r_t$ fixed observations or are they learnable and also backpropagated through to the prediction values $y$ and consequently through the prediction head?**
> >
> > Our loss is not static but is dynamically calculated based on the pairs suggested by the query head during each optimization step (which means the histories $H_t$ in each step are different). PABBO employs policy gradient reinforcement learning to train the query head. The key idea is that the reward signal is used to directly optimize the policy, allowing the query head to adaptively suggest query pairs that maximize cumulative utility over the entire trajectory. The reward values $r_t$ are not fixed, but are dynamically computed based on the utility of the query pairs selected at each step. Reward values are not backpropagated through the prediction values $y$. Instead, $r_t$ serves as a signal for optimizing the query head using policy gradient.
> >
> > - **This step looks very similar to offline BO where a offline data set is used to learn an acquisition function or a function approximation. What is the relationship of this method to offline BO?**
> >
> > We believe amortized optimization, a class of methods PABBO belongs to, differs from offline BO. In offline BO, an existing dataset is used to train a surrogate model or acquisition function. In contrast, our method actively collects data through the query head, which generates query pairs during the optimization process. The dynamic nature of this process allows the model to iteratively improve its understanding of the optimization space, while offline BO relies solely on pre-existing data without adaptive query generation.
> >
> > **Line 303, How is the query set constructed when computing $\pi_\theta$
> > ? Is it a random subset of all query pairs? Or is any specialized strategy used to locate the most promising queries?**
> >
> > During pre-training, for each task, we optimize the policy's decision on a set of $M$ query pairs. The reviewer is correct,  this set is a random subset of all query pairs from the possible combinations of $S$ query points. This strategy is simple but also explorative while controlling computational cost: we want the policy to see as varied locations as possible (more query points) during pre-training, but it is computationally heavy to query all the possible pairs, thus, we only use a subset. At test time, however, all pairs will be used to take all query candidates into account. Further information can be found in the comment answer 2/2 made to reviewer Tep5.
> >
> >
> > **How is PABBO able to avoid sub-optimal solutions caused due to not exploring the whole optimization space?**
> >
> > As mentioned previously in the last question, we randomly sample query points across the design domain, to make sure the policy sees many possible locations during meta-training. At test time, we discretize the optimization space with $S$ quasi-random query points and pass *all* the possible combinations to the policy.
> >
> > **The full potential of the algorithm is only realized in large experiment with thousands of observations. Such large scale experiments have not been presented in the paper.**
> >
> > We would like to emphasize that the primary goal of PABBO is to facilitate user preference optimization. Our method trades off a long but offline pre-training time with 1) fast inference time and 2) improved accuracy. Both these advances reduce the overall time the user has to spend interacting. Indeed, we inherently assume that the user's time is extremely valuable, and as such, cases where several hundred of sequential interactions are necessary do not seem plausible, specifically given that the results would be hampered by fatigue.

---

### Official Review · Reviewer_Xx1d · 2024-11-06

**Soundness:** 3
**Presentation:** 4
**Contribution:** 3
**Rating:** 8
**Confidence:** 4

**Summary:**

This paper explores preferential black-box optimization with an amortized inner-loop solver. In other words, RLHF but from the lens of Bayesian optimization instead of NLP.

**Strengths:**

This work is well motivated. Aligning training objectives with downstream use makes sense (i.e. the explicit acquisition head) and amortization also makes sense if you expect to solve the same (or nearly the same) optimization problem many many times.

The paper is clear and well-written

The experiments hit pretty much all the key points I would expect

**Weaknesses:**

This paper seems to treat simple and cumulative regret as interchangeable, although I am sure the authors know the difference. If the policy is optimized to minimize cumulative regret, why is cumulative regret not reported? On the other hand if simple regret is really what you care about, why are none of the baselines aimed at best-arm identification? For example, Thompson sampling is optimal w.r.t. cumulative regret, not simple regret. Top-two thompson sampling is optimal in the discrete case [1], although translating to the continuous case requires some thought (what epsilon is sufficiently large to constitute a different arm?).

I also feel that the experimental design could be more directed at identifying under what conditions this solution makes sense. What is the breakpoint when amortization starts paying off compared to direct search at test time? How far can you push this before it breaks, especially in terms of the initial data package?

It also seems very odd that preference tuning for language models is not mentioned even once, even though it is obviously the same problem and basically the same solution (ignoring low-level implementation choices). I can't tell if this is a deliberate choice that reflects the legal sensitivity around LLMs right now or a genuine oversight, however I see no reason not to draw the connection to preference tuning work.

I generally try to avoid policing language, but sentences like "... amortization ... has emerged as an almost magical bullet solution" clearly crosses the line from excitement to needless hype. Amortization is not anything close to a magic bullet. You know that and I know that. It is just moving compute around from test time to train time. Of course science involves some marketing, but language and framing like this hurts our credibility with more staid scientific disciplines, people that you presumably would like to take you seriously. Write like a scientist, not a salesman. It's ok to be excited, just don't get carried away!


References

- [1] Russo, D. (2016, June). Simple bayesian algorithms for best arm identification. In Conference on Learning Theory (pp. 1417-1418). PMLR.

**Questions:**

Can you add a reasonable interpretation of top-two thompson sampling in the continuous case to your experiments by the end of the rebuttal period?

Are you warm-starting the amortized solver training as the data changes? Do you lose performance because of that choice? How expensive does the test-time search have to be to warrant amortization? How carefully have you thought about the FLOPS involved here?

How much can you push this? I don't just want to see rosy results, I want to know when this breaks.

---

> ### Author Response · Authors · 2024-11-19
> **Answer 1/2**
>
> First, we would like to point to the global comment we posted, which tackles several remarks made by the reviewer.
>
> **This paper seems to treat simple and cumulative regret as interchangeable, although I am sure the authors know the difference. If the policy is optimized to minimize cumulative regret, why is cumulative regret not reported? On the other hand if simple regret is really what you care about, why are none of the baselines aimed at best-arm identification? For example, Thompson sampling is optimal w.r.t. cumulative regret, not simple regret. Top-two thompson sampling is optimal in the discrete case, although translating to the continuous case requires some thought (what epsilon is sufficiently large to constitute a different arm?).**
>
> Answer:
>
> **About Thompson sampling:**
>
> In our experiments, we evaluated PABBO against a Gaussian process-based acquisition strategy, $q$TS. This baseline was originally proposed by [1] and subsequently used by [2] in their experiments. $q$TS is a straightforward application of a *batch* BO acquisition function to the PBO case: for $q=2$, the strategy samples two continuous draws from the posterior distribution of the GP, optimizes each draw, and yields a batch containing the two maximizers. Translating this AF to the PBO setting, preferential feedback is obtained by comparing the two queries of the batch.
> As noted by [2], adapting batch BO strategies to the PBO setting remains heuristic, as such strategies do not acknowledge that the optimized designs are meant to be compared. Still, the diversity provided by maximizing two different draws of the posterior might yield informative comparisons, which is why this baseline remains used in current benchmarks. As a matter of fact, recent work established that Dueling Thompson sampling ($q$TS with $q=2$) is asymptotically consistent, for a finite query set [3]. While this result is not as strong as a regret bound, it provides some mathematical justification for this strategy. Additionally, regarding best-arm identification the GP-based strategy qEUBO [2], which we compare against, is one-step Bayes optimal.
>
> With all that said, we emphasize that applied to PBO, $q$TS is *not* a top-two Thompson sampling algorithm, which favors exploration by randomizing among two candidate arms. We agree with the reviewer that
> contrarily to ''vanilla Thompson sampling'', top-two Thompson sampling is optimal in terms of simple regret in the discrete case, due to the added exploration.
> Additionally, we apologize for the confusion and warmly thank the reviewer for pointing us toward this quite interesting literature, which was unknown to us. **We will clarify how the baseline $q$TS works in the updated PDF.** As a side note, this discussion raises the point of whether we *should* consider an additional baseline actually implementing top-two Thompson sampling in the preferential black-box optimization setting, using a discrete query set. This would imply randomizing among two candidate *pairs*. While the idea sounds interesting, it is not a priority for this work.
>
> **About cumulative and simple regret:**
>
> During pre-training, we use the (discounted) cumulative simple regret as a reward (Equation 3, main text), since it sends a stronger and less sparse signal than simple regret. Now, at inference time, when encountering a new task, we report simple regret. This is because the use cases we have in mind for PABBO are human preferences optimization. For instance, we might want to find the best sushi product for a given user, or the best visual design for a website, tasks where typically the notion of cumulative regret matters less than simple regret. **We added a sentence in the PDF to justify our choice of the cumulative simple regret as a training reward, and we will provide cumulative regret plots for some of the experiments carried out, in the Appendix.**
>
> [1] Siivola et al., Preferential batch Bayesian optimization
>
> [2] Astudillo et al., qEUBO: A Decision-Theoretic Acquisition Function for Preferential Bayesian Optimization
>
> [3] Astudillo et al., Preferential Multi-Objective Bayesian Optimization

---

> > ### Author Response · Authors · 2024-11-19
> > **Answer 2/2**
> >
> > **Are you warm-starting the amortized solver training as the data changes? Do you lose performance because of that choice? How expensive does the test-time search have to be to warrant amortization?
> > How carefully have you thought about the FLOPS involved here?**
> >
> > - If we understood the question correctly, at present, PABBO only needs to be pre-trained once when applied to diverse tasks of the same dimensionality. Tasks with different dimensionality will require pre-training PABBO from scratch again.
> > - In human-interactive application scenarios, the decision of whether amortization is warranted ultimately depends on the user's tolerance for computation time. This subjective threshold is determined by how long users are willing to wait for an interactive response.
> >     To that end,  the whole purpose of PABBO is to make learning from human feedback possible in practice. We chose to rely on preferential feedback and to make test-time interaction rounds with users as fast as possible. An objective answer would require defining a precise threshold on what users consider ''fast'' interaction time.
> > - Nevertheless, if we can handle a long but offline optimization, this tradeoff does not even matter anymore, given that PABBO provides *better* solutions in *faster* time. As the reviewer points out, the only thing left to consider might be the overall ''cost'' associated with pre-training. On $2$-dimensional problems, the pre-training step of PABBO amounts to 20 hours on a GPU. Therefore, to justify our approach, we implicitly assumed that 20 hours of training on a GPU compares favorably with the ''cost'' reduction yielded by our methods, due to reduced user waiting times.
> > Importantly, given that the model can be used to learn the preferences of multiple independent users, we consider that the above mentioned assumption is reasonable.
> > ''cost'' is here to be understood in a broad sense: electric power, time used for computations, time spent waiting for a user, etc.
> >
> > **How much can you push this? I don't just want to see rosy results, I want to know when this breaks.**
> >
> > At present, our main limitation is the size of the query set. We refer to the answer made as a comment to reviewer Tep5.

---

> > > ### Comment · Reviewer_Xx1d · 2024-11-20
> > > **response acknowleged**
> > >
> > > thanks for your response!

---

### Official Review · Reviewer_Tep5 · 2024-11-07

**Soundness:** 3
**Presentation:** 4
**Contribution:** 3
**Rating:** 8
**Confidence:** 3

**Summary:**

This work introduces a novel approach to Preferential Bayesian Optimization (PBO) by fully amortizing the optimization process. Traditional PBO methods rely on GPs for modeling user preferences between design pairs, requiring extensive computational resources due to approximate inference for non-conjugate likelihoods. PABBO addresses this challenge by employing a transformer-based neural architecture and reinforcement learning to learn both the surrogate model and acquisition function end-to-end, significantly accelerating the optimization process. By pre-training on synthetic and real-world datasets, PABBO achieves several orders of magnitude in speed improvement while often surpassing GP-based methods in optimization quality.

**Strengths:**

- The paper is well written and easy to understand for the most part.
- The proposed method is a novel and interesting application of amortized learning/BO for PBO setting. The empirical results seem superior than traditional PBO methods and acquisition functions.
- Nice set of experimental evaluation and ablation studies — though I’d love to see a more comprehensive experimentation section (see weaknesses below) for methods that are hard to provide theoretical guarantees.

**Weaknesses:**

- Evaluation on harder problems are very limited. Most test functions are of very small dimensions and may not resemble real-world optimization tasks. The only experiments with moderate dimensionalities are 6 and 9-dims respectively and are all from the HPO-B. This leaves the scalability of the proposed methods in question.
- No ablation studies on meta-learning training set. It is plausible that amortized learning methods are highly sensitive to the selection of pre-training data. There’s no justification or ablation on the choice of pre-training data for PABBO and it’s unclear how robust the method is to a different set of pre-training data.
- The performance advantage vs baseline methods are not significant in many cases.
- While it is true that estimating the true posterior of a preference model’s posterior might be expensive, variational inference or Laplace approximation with some clever implementation can go quite far and allows for gradient-based optimization for acqf value. On the other hand, for larger models like PABBO1024, the inference time advantage of amortized learning might be diminishing when compared to GP-based model according Figure 3 and 5.

**Questions:**

- S in query set is from a Sobol sample. How big is sufficient? How would this affect the optimization quality as even the largest experimented S=1024 might not be sufficiently for moderately high-dimensional spaces, limiting the scalability of this PABBO.
- How does PABBO work for higher dimensional problems? All experiments presented are of very small dimensions and may not resemble real-world optimization tasks.
- Algorithm 1 seems to only describe the meta-learning/pre-training part of PABBO as we are still updating the PABBO model at the end of the loop. What’d be the algorithm for applying PABBO on new, unseen dataset? I’d assume that’d be the same as the inner loop of the algo, but it’d be nice to have some clarification from the author.

---

> ### Author Response · Authors · 2024-11-19
> **Answer 1/2**
>
> Comments that can be answered through additional experiments will be posted as soon as possible when the latter are completed.
> With respect to the reviewer's comments, these include:
>
> - Consider higher dimensional examples.
> - Perform an ablation study on the composition pre-training dataset.
>
> Additional comments can be found in the global comment.
>
> **The performance advantage vs baseline methods are not significant in many cases.**
>
> While we do not always reach a clear statistical significance, our proposed method consistently ranks among top performers in every experiment. This consistency is a key takeaway, as it suggests that once pre-trained, we can effortlessly apply PABBO to a wide range of problems, whereas other methods require specifying a kernel and an acquisition function, choices that are typically heavily task-dependent and need to be thought thoroughly.  We speculate that the versatility of our algorithm stems from the rich dataset used for pre-training, combined with the fact that we are learning the acquisition function from data.
>
> **While it is true that estimating the true posterior of a preference model’s posterior might be expensive, variational inference or Laplace approximation with some clever implementation can go quite far and allows for gradient-based optimization for acqf value. On the other hand, for larger models like PABBO1024, the inference time advantage of amortized learning might be diminishing when compared to GP-based model according to Figure 3 and 5.**
>
> We agree with the reviewer. However, in our experiments, the GP baselines do employ Laplace approximation for posterior approximation. The implementation is that of BoTorch [1], a state-of-the-art library for GP-based modeling and BO. Even though we did not carefully check that the implementation is ''as fast as possible'', we remain confident that the issue is mitigated, from this side. Regarding PABBO, we acknowledge that larger models like PABBO1024 require longer inference time as seen in Figures 3 and 5. However, an important distinction arises when scaling the tasks (e.g., increased dimensionality or number of iterations). The inference time of PABBO grows linearly with the size of the task, whereas GP-based methods typically exhibit superlinear growth with respect to task dimensionality. This makes PABBO particularly advantageous in scenarios with many iterations, where the gap in total runtime between PABBO and GP-based methods becomes increasingly significant.
>
> [1] Balandat et al. BoTorch: A framework for efficient Monte-Carlo Bayesian optimization.
>
> **Algorithm 1 seems to only describe the meta-learning/pre-training part of PABBO as we are still updating the PABBO model at the end of the loop. What’d be the algorithm for applying PABBO on new, unseen dataset? I’d assume that’d be the same as the inner loop of the algo, but it’d be nice to have some clarification from the author.**
>
> The reviewer is correct, we forgot to include an algorithm description of how PABBO operates at test time. **This is now corrected and constitutes an additional section of the Appendix.**

---

> > ### Author Response · Authors · 2024-11-19
> > **Answer 2/2**
> >
> > **S in query set is from a Sobol sample. How big is sufficient? How would this affect the optimization quality as even the largest experimented S=1024 might not be sufficiently for moderately high-dimensional spaces, limiting the scalability of this PABBO.**
> >
> > We appreciate the reviewer for raising this insightful question. In our experiments, ranging from 1- to 9-dimensional spaces, we found that $S=1024$ was sufficient to achieve a remarkable inference time/accuracy trade-off. The very purpose of our algorithm is to facilitate sequential learning of human preferences, by reducing the waiting time for users, making it virtually nonexistent. Since humans do not excel at comparing high-dimensional objects, the typical use case for PABBO is low-dimensional.
> >
> > Nevertheless, we acknowledge that high-dimensional settings constitute a general challenge for any approach that relies on discretized candidate points. While directly increasing the size of $S$ is one straightforward solution, this comes at the expense of PABBO's efficiency (see Figure 5c). We now elaborate on several promising solutions to improve the scalability of our method:
> >
> > - **Direct optimization of the output values:** Since the output values of PABBO (acquisition scores) are differentiable with respect to the input query pairs, one potential direction is to directly optimize the acquisition function via gradient descent. This would allow us to propagate gradients back to the input space, potentially identifying high-quality query pairs without requiring a fixed discretization. GP-based strategies follow the same principle.
> >
> > - **Deterministic policy:** Another direction is to develop a deterministic policy that directly outputs optimal query pairs instead of relying on a pre-sampled query set. This technique would be similar in spirit to the pairwise extension of Deep Adaptive Design [2], focusing on generating highly informative pairs in each iteration. While promising, this method introduces additional challenges in designing and training such a policy.
> >
> > - **Iterative query set updates:** An alternative approach is to iteratively refine the query set, in line with the field of local Bayesian Optimization [3]. For example, after a few rounds of optimization, the high-potential regions identified from previous iterations could guide the next sampling phase, focusing only on these promising areas. This would reduce the effective search space and allow for more efficient optimization.
> >
> > [2] Adam et al. Deep Adaptive Design: Amortizing Sequential Bayesian Experimental Design.
> >
> > [3] Eriksson et al. Scalable Global Optimization via Local Bayesian Optimization.

---

> > > ### Comment · Reviewer_Tep5 · 2024-11-22
> > >
> > > I thank the author for their response and look forward to their additional experimental results. While I'm convinced by some of the author's response, with regard to other responses:
> > >
> > > > Limitation on higher-d problem is a general challenge for BO
> > >
> > > While high-d optimization itself is a challenging task for vanilla BO and as the author rightfully noted, there are many dedicated work investigating methods in this space. I believe PABBO suffers more from this issue compared to other BO methods.
> > >
> > > Given a query set S, it'd require the transformer to run inference over S^2 pairs, which is over one million pairs for |S| = 1024. It is indeed a fundamental limitation on methods based on  discretized candidate points, but methods based on smaller models such as a GP can afford performing optimization (with the reparameteriztaion trick as BoTorch is using) directly over a 2d space for d-dimensional query points, while it will be challenging to do the same on a reasonably large transformer model.
> > >
> > > While I acknowledge that it is not necessary to solve every single question around the proposed method to get a paper pass the acceptance bar, the scalability issue of PABBO seems rather fundamental and the paper can benefit from a more thorough discussion of it as well as some of the direction to mitigate such limitation as the author has mentioned in the response.
> > >
> > > > Despite no stats-sig experimental results, PABBO is consistently directionally better. Baseline performance.
> > >
> > > While it’s nice to see consistent directional results, one easy patch is to run more replications to narrow down the CI. Currently for many test problems, there are essentially no significant difference between all methods.
> > >
> > > Additionally, in the qEUBO paper (Astudillo et al, 2023), qEUBO clearly outperformed qTS while this paper concludes otherwise in many test problems. One possibility is that the qEUBO paper uses a [VI implementation](https://github.com/facebookresearch/qEUBO/blob/main/src/models/variational_preferential_gp.py) of preferential GP while the author uses the laplace approximiation of PairwiseGP, which could be anti-conservative --- I wonder if this might explain some of the performance gap between qEUBO and and PABBO. While I sympanthize with the author on the tight timeline, it could be helpful to better understand whether the performance gap is coming from the method itself or is an artifact of a GP model with poorly quantified uncertainty, if the time allows.
> > >
> > >
> > > Despite the limitation mentioned, this paper is overall an interesting exploration of amortized learning in the space of preferential BO. I thank the author for making the effort to work on additional experiments within this limited discussion period and I will reconsider my evaluation upon seeing the promised empirical results.

---

> > > > ### Author Response · Authors · 2024-11-23
> > > > **Responses related to additional experiments and further questions**
> > > >
> > > > We thank the reviewer for their solicitude. While scalability is a crucial issue, we see this concern as being methodological, rather than practical: as mentioned above, when interacting with users, high-dimensional comparisons seem difficult to handle already.
> > > >
> > > > Nevertheless, we believe the reviewer raised a legitimate concern.
> > > >
> > > > **We have included an additional experiment on a 16-dimensional HPO-B task, with results presented in Figure 10 of the updated PDF. PABBO still significantly outperforms all other baselines in terms of simple regret on this task.**
> > > >
> > > > Regarding the scalability concerns at higher dimensions due to the query set size, we acknowledge that this is a current limitation. As discussed in our previous responses, this issue could potentially be addressed through a series of proposed solutions. Additionally, we propose a more practical approach to mitigate this limitation through parallel computations. Currently, a query set of size $S$ is passed to PABBO. We can instead pass a batch of $B$ query sets of size $S$, and choose the next candidate pair from the entire batch. This increases the effective query set size while further speeding up inference times via parallelization, and resulting in faster convergence. The results are shown in Figure 12, where, using $5$ batches of size $S=256$, we favorably compare to the PABBO version that solely uses $1$ batch of size $S=1024$, in a reduced inference time. Finally, another promising avenue for work regarding the high dimensional setting is to consider a bi-dimensional transformer architecture that eliminates the fixed input dimensionality constraint. Building on recent work on the matter, one could leverage low-dimensional tasks to potentially accelerate convergence on high-dimensional tasks. This is typically useful if the objective possesses a hidden low-dimensional structure, or decomposes additively (e.g., the Ackley $D$-dimensional function writes as a sum).
> > > >
> > > > Regarding the potential **sensitivity of the composition of the pre-training dataset**, we appreciate the reviewer’s comment and agree that this is a key factor in training an effective PABBO. This is now addressed in a dedicated ablation study. Specifically, we compared the performance of PABBO pre-trained on GP draws sampled exclusively from an RBF kernel to the original approach, which uses a mixture of kernels (Matern and RBF). The results, shown in Figure 6, meet our expectation: pre-training on a mixture of kernels yields better performance than using only the RBF kernel. This improvement is likely due to the increased diversity of synthetic samples the mixture provides, which improves the model's generalization capability.
> > > >
> > > > Finally, regarding the results between qEUBO and qTS, the above-mentioned experiment conducted in Figure 12, the Ackley 6D function, is one of the tasks used in the qEUBO paper. For this test function, similar to the qEUBO paper, our results show that qEUBO outperforms qTS. This being said, as pointed out by the reviewer, different inference schemes for GP posterior inference may yield different results. This phenomenon was further studied by the authors of [1]. The use of VI in the original qEUBO paper, compared to the Laplace approximation used in BoTorch implementation, could contribute to the observed performance differences. We acknowledge this as a potential factor worth exploring further when time permits.
> > > >
> > > > Thank you again for your thoughtful and constructive comments, we deeply appreciate them!
> > > >
> > > > [1] Takeno et al. Towards Practical Preferential Bayesian Optimization with Skew Gaussian Processes.

---

> > > > > ### Comment · Reviewer_Tep5 · 2024-11-26
> > > > >
> > > > > I thank the author for their response and additional experimental result. While I'm not fully convinced by some of the author's response, I do think this work could be an interesting addition to the field and have raised my score (with a lower confidence score).
> > > > >
> > > > > In the final version of the paper, I expect the author can include a more comprehensive discussion on PABBO's limitation and potential future directions as well as refined experimental results & interpretation, especially the newly added ones (e.g., in A.2.1, why PABBO has higher cumulative results than PABBO RBF? Can we try a more diverse set of training data mix?)

---

> > > > > > ### Author Response · Authors · 2024-11-27
> > > > > > **Additional response: adressing limitations in discussion and rectification on pre-training ablation study**
> > > > > >
> > > > > > Thank you for raising the score and for your valuable feedback. We sincerely appreciate your
> > > > > > thoughtful consideration of our response.
> > > > > >
> > > > > > Regarding your suggestion about limitations and future
> > > > > > directions: **The discussion (Section 6) now features an additional paragraph discussing PABBO’s
> > > > > > limitations on the query set size S scaling with task dimensionality**.
> > > > > >
> > > > > > Concerning the inconsistency in
> > > > > > cumulative regret results (A.2.1): We sincerely apologize for any confusion. Upon careful review, we
> > > > > > discovered a code error when calculating cumulative regret for PABBO RBF. This oversight occurred
> > > > > > due to the rapid implementation of multiple experiments during the rebuttal period. We have now
> > > > > > fixed this error and will ensure all rebuttal experiments undergo thorough verification. Furthermore,
> > > > > > to ensure full transparency and reproducibility, **we provided a minimal code for reproducing Figure 6
> > > > > > in the supplementary material**. We thank the reviewer for spotting this problem.
> > > > > > The final version will further include a more comprehensive analysis of different pre-training data
> > > > > > compositions and their effects on performance.

---

### Author Response · Authors · 2024-11-19

We thank the reviewers for their careful consideration and the time they dedicated to our submission. The reviewers agreed on the following strengths of the paper:

**Methodology:** ''The proposed method is a novel and interesting application of amortized learning/BO for PBO setting'' (Tep5, x3e8), ''The proposed approach is intuitive and a reasonable way to model preferences in BO'' (dRx5). ''Aligning training objectives with downstream use makes sense  and amortization also makes sense if you expect to solve the same (or nearly the same) optimization problem many many times.'' (Xx1d).

**Relevance:** ''This work is well-motivated'' (Xx1d).

**Empirical evaluation:** ''The experiments hit pretty much all the key points I would expect'' (Xx1d),
''Nice set of experimental evaluation and ablation studies`` (Tep5),
''Extensive experimentation shows that PABBO has a smaller inference cost while achieving a smaller regret compared to prior approaches''. (dRx5, x3e8)

**Clarity/Presentation:** ''well-written'' (Tep5, Xx1d),  ''easy to understand for the most part'' (Tep5).

--------

**Additional experiments:**

All reviewers made remarks that can be answered partly by carrying out further experiments. These include but are not limited to, ablation studies related to PABBO's hyperparameters, high-dimensional settings, higher number of iterations, and varying the composition of the pre-training dataset.
We will try to carry out all experiments in the allocated time, and we thank the reviewers for giving us actionable insights. These can only improve the current submission.

This being said, due to limited computational resources, some performing additional experiments takes time, and some of them might not terminate in time. In the interest of making the most of the discussion period, we would still like to engage in a fruitful discussion with the reviewers. Therefore, we begin by answering the questions that do not require additional experiments in separate comments.

--------

**Updated PDF of the submission:**

Finally, as authorized by ICLR, we are currently preparing a revised PDF and will let you know as soon as it is fully ready.
Currently, the PDF includes an additional algorithm describing how PABBO operates at *test-time*, as suggested by Reviewer Tep5. (Appendix A.1), and a justification for why we mainly report single regret during the experiments. Additionally, The PDF will:

- Contain the additional experiments performed, in a dedicated section in the appendix.
- Include an extended related work section in the appendix, focusing on preference tuning for large language models. While this area was not a primary focus of our study, we recognize its relevance to our research and its potential to inspire extensions of our method, and we thank the reviewer Xx1d for pointing us toward this literature.
- Elaborate on the embeddings used by PABBO, specifically the use of component-wise addition.
- Add cumulative regret plot, as suggested by reviewer Xx1d, in the appendix.
- Fix the typos identified by Reviewer do2x, and generally employ a more measured tone, as suggested by Reviewer Xx1d.

---

### Author Response · Authors · 2024-11-23
**Global Comment 2 - Updated PDF and experiments**

We have now edited the PDF. **Changes are highlighted in red**. They are either from reviewer requests or can be seen as the output of the engaged discussions. Furthermore, multiple experiments requested by reviewers have been added to the appendix. For commenting on the results, we respond in individual comments. Overall, changes include:

- A more measured tone regarding the sentence L45 as suggested by (Xx1d).

- Definition of $y_{i,1}$ and $y_{i,2}$ L189 (do2x).

- Justification on why using a Gaussian Likelihood for Equation 4 L277 (do2x).

- An algorithm to describe how PABBO operates at test-time, presented in Appendix A.1.1
and referred to in the main text at L299.

- A paragraph of related work on preference tuning for language models L351 (Xx1d).

- A brief description of how the baseline qTS works L376 (Xx1d).

- A sentence elaborating on the fact that while we use (discounted) cumulative regret during pre-training, our main interest is simple regret, although for completeness, both are reported. L402 (Xx1d).

- A supplementary section Appendix A.1.2 providing additional justifications on our transformer architecture, namely the use of three separate embeddings for $x_{i,1}, x_{i,2}$ and $l_i$, with a component-wise-sum to obtain the token embedding for $x_{i,1},x_{i,2},l_i$ (dRx5).

- An additional experiment on the composition of the pre-training dataset in Appendix A.2.1, Figure 6 (Tep5).

- Cumulative simple regret plots for all test functions in Appendix A.2.2, Figure 7 (Xx1d).

- An additional experiment studying the impact of the loss hyperparameter $\lambda$, controlling how important the prediction task should be, in Appendix A.2.3, Figure 8 (do2x).

- An additional experiment on the discount factor $\gamma$ that uses a larger grid for $\gamma$, and also reports cumulative simple regret, in Appendix A.2.4, Figure 9 (do2x).

- A higher dimensional experiment using HPO bench and search space No. 5971, which is $16$-dimensional, in Appendix A.2.5, Figure 10 (Tep5).

- The addition of another baseline, Multinomial Predictive Entropy Search (MPES), and its evaluation on synthetic examples, presented in Appendix A.2.6, Figure 11 (x3e8).

- An additional test function, the 6-Dimensional Ackley function. In particular, for this example, to address the concerns regarding query set size scalability (Tep5), we include a ''batch'' version of PABBO. This experiment is available in Appendix A.2.7, Figure 12 (x3e8).

- An additional experiment on the query budget $T$ which considers different values of $T$, and also compares two versions of PABBO: with or without giving $t/T$ as an input to the acquisition head (see Figure 2 right). This experiment is available in Appendix A.2.8, Figure 13 (do2x).

- An additional experiment on the percentage of $D^{\text{ctxpred}}$ and $D^{\text{tarpred}}$. We fix the $D^{\text{ctxpred}}$ and run the prediction task with different $D^{\text{tarpred}}$. This experiment is available in Appendix A.2.9, Figure 14 (do2x).

---

### Meta-Review · Area_Chair_r8i6 · 2024-12-28

**Metareview:**

This work proposes an approach to generating amortized policies for preferential Bayesian optimization (PBO) using transformer neural processes and reinforcement learning.  In their experiments, they appear to be able to achieve parity with established methods for PBO.  The meta-training procedure allows the authors to pre-train off of relevant priors which can lead to improved sample complexity of the algorithms (e.g., meta-tasks from HPO-B benchmarks).

Reviewers were overall quite positive, finding the paper to be well-written, clearly motivated, and well-executed.

All reviewers expressed some amount of skepticism of the results that could be addressed via more rigorous experiments and modest framing.  This is interesting work that will be published and read, so the authors have the opportunity to demonstrate what the limitations are (Xx1d). Exposing such limitations will ultimately benefit the community and will help stimulate future directions for this line of work.

Some reviewers expressed concerns that runtime improvements from PABBO may be overstated (Tep5) (e.g., S=1024 appears necessary, for the highD problems, but illustrations of runtime comparisons in the MT focus on the case of S=256 (Fig3)—which is orders of magnitude faster than S=1024 (Fig 5c)).   Reviewer Xx1d asked for further contextualization of the runtimes.

There are a few remaining gaps.  First, the authors do not run enough replicates to demonstrate whether certain methods are statistically significantly better than others (see cf., review+discussion w/ Tep5).  In fact, it’s somewhat ambiguous what statistics are reported in the figures: are the authors plotting ± 1 standard deviation, or ±1 std of the mean (standard error of the mean, SEM).  It is common practice to report the 95% CI for the mean (i.e., 1.96*std/sqrt(num replications)).  Please update your figures to use at least 30 replicates and report 1.96*SEM for the intervals.

Reviewer Tep5 also asked about sensitivity to the pre-training and whether the advantages of PAABO in terms of sample complexity/runtime would hold if alternate models (such as the pairwise kernel+VI from Astudillo et al.’s qEUBO paper) were used.  The responses from the authors were not satisfactory to the reviewer (as well as the AC).  Addressing these points in the CR will strengthen the work.

I have also read the paper in some detail, a few brief comments:
- The updated PDF does not adequately explain how qTS works.  It sounds like you are optimizing the sample path, vs doing discrete sampling (say, using the same S used by PAABO).  Please clarify the exact procedure, and if optimizing sample paths, describe what method you are using (e.g., RFFs, pathwise conditioning, etc).
- Why is only S=1024 used in the higherD (HPO-B) examples in Fig4, but it is evaluated with lower cardinality S for the lowerD examples (Candy/Sushi).  The authors discuss how larger S might be necessary for higherD problems (see comments from Tep5). Including these lower S’s in this plot, or having a more in-depth ablation could help make this point better).  Reporting the wall time of the methods here would also be instructive, since the wall time improvements over GPs may be much thinner with S=1024.
- The superior sample complexity of PAABO is by far most apparent on the HPO-B datasets where (to the best of my understanding), PAABO the model / policy is pre-trained on the meta-task itself.  Does PAABO’s improvement lie in learning a better policy, or having a better prior?  Fig6 from the rebuttal does not address Reviewer Tep5’s point.  Running the HPO-B benchmarks with the standard prior used for the synthetic functions would address this question.
- The motivation for the HPO-B datasets is not very clear, since it's not obvious why one would want to do PBO with HPO datasets.  Some additional discussion here would be helpful.  It would also be nice to see more standard or real-world motivated PBO problems from previous works (Tep5 gives some suggestions).

**Additional Comments On Reviewer Discussion:**

The authors and reviewers engaged in a discussion, with a few additional clarifications and experiments conducted.  These experiments did not temper the skepticisms of reviewers per se, but as seen in the scores—the paper is interesting and well-executed enough to make it in.

---

### Decision · Program_Chairs · 2025-01-22

Accept (Spotlight)